# Iron-Based Ceramic Composite Nanomaterials for Magnetic Fluid Hyperthermia and Drug Delivery

**DOI:** 10.3390/pharmaceutics14122584

**Published:** 2022-11-24

**Authors:** Ming-Hsien Chan, Chien-Hsiu Li, Yu-Chan Chang, Michael Hsiao

**Affiliations:** 1Genomics Research Center, Academia Sinica, Taipei 115, Taiwan; 2Department of Biomedical Imaging and Radiological Sciences, National Yang Ming Chiao Tung University, Taipei 112, Taiwan; 3Department and Graduate Institute of Veterinary Medicine, School of Veterinary Medicine, National Taiwan University, Taipei 106, Taiwan

**Keywords:** iron-based nanoparticles, ceramic nanocomposites, magnetic resonance imaging, magnetic fluid hyperthermia, drug delivery

## Abstract

Because of the unique physicochemical properties of magnetic iron-based nanoparticles, such as superparamagnetism, high saturation magnetization, and high effective surface area, they have been applied in biomedical fields such as diagnostic imaging, disease treatment, and biochemical separation. Iron-based nanoparticles have been used in magnetic resonance imaging (MRI) to produce clearer and more detailed images, and they have therapeutic applications in magnetic fluid hyperthermia (MFH). In recent years, researchers have used clay minerals, such as ceramic materials with iron-based nanoparticles, to construct nanocomposite materials with enhanced saturation, magnetization, and thermal effects. Owing to their unique structure and large specific surface area, iron-based nanoparticles can be homogenized by adding different proportions of ceramic minerals before and after modification to enhance saturation magnetization. In this review, we assess the potential to improve the magnetic properties of iron-based nanoparticles and in the preparation of multifunctional composite materials through their combination with ceramic materials. We demonstrate the potential of ferromagnetic enhancement and multifunctional composite materials for MRI diagnosis, drug delivery, MFH therapy, and cellular imaging applications.

## 1. Introduction

Magnetic materials are functional materials with great potential that are widely used in biomedicine [1,2,3]. Their unique magnetic signals allow them to be used as sensors in imaging medicine, based on the detection of geomagnetic fields, and in noncontact magnetic-field-heating therapy [4,5,6]. Magnetic materials can even integrate all the conditions of nanoparticles when their particle size is limited to within a range of 1–100 nm. Nanoparticles can be used as contrast agents, target drug carriers, and multifunctional magnetic biomedical materials for controlled and focused therapy. In recent years, nanoparticles have been compounded with other materials possessing low toxicity and superparamagnetic and biocompatible properties. The resulting compounds can be applied for using in similar applications as for nanoparticles mentioned earlier [7,8,9]. Once nanosized, these nanomaterials exhibit many novel and excellent properties [10]. The field of nano-biomedical materials involves the integration of nanomaterials/nanotechnology with biomedical materials or drugs, and these developments have substantially contributed to the progress of human medicine.

In all mammalian cells, iron is an indispensable element for the processes of cell growth and differentiation. Because of the unique physicochemical properties of magnetic iron-based nanoparticles, such as superparamagnetism, high saturation magnetization, and high effective surface area, they have been applied in biomedical fields such as diagnostic imaging, disease treatment, and biochemical separation [11]. Iron-based nanoparticles have been used in magnetic resonance imaging (MRI) to produce clearer and more detailed images, and they have therapeutic applications in magnetic fluid hyperthermia (MFH) [12,13]. The combination of the treatment and diagnosis approach into one system for cancer treatment indicates their potential for ushering in the “iron age”. In recent years, clay minerals, such as ceramic materials with iron-based nanoparticles, have been used to construct nanocomposite materials to enhance their saturation magnetization and thermal effects. Due to their unique structure and large specific surface area, iron-based nanoparticles can be homogenized by adding different proportions of ceramic minerals before and after modification to enhance saturation magnetization. In this review, we assess the potential to improve the magnetic properties of iron-based nanoparticles and in the preparation of multifunctional composite materials through their combination with ceramic materials. We demonstrate the potential of ferromagnetic enhancement and multifunctional composite materials for MRI diagnosis, drug delivery, MFH therapy, and cellular imaging applications [14].

Multifunctional nanocomposites have been a hot area of research in recent years. In this review, we hoped to identify iron-based nanocomposite materials that can enhance saturation magnetization (Ms) and be applied to optimize MRI contrast. In addition, a magnetic nanocomposite material with improved biocompatibility is needed for biomedical applications. As for the choice of the composite material, examples of low-cost, high-adsorption, and biocompatible ceramic materials, montmorillonite silicate, kaolinite minerals, or bioglass can be used to produce multifunctional nanocomposite materials with both magnetic properties and high adsorption performance.

## 2. Magnetic Properties of Nanoparticles

Nanotechnology has become ubiquitous in everyday life through its use in the aerospace, electronic, cosmetic, and pharmaceutical industries, among others. These developments have enabled improvement of the existing nanoparticle properties and the introduction of new optical, electrical, and mechanical functions. In addition, nanosized materials experience small size, surface, quantum tunneling, Coulomb blocking, and quantum-limiting effects distinctly from macroscopic materials.

Hence, the optical, thermal, and electrical effects as well as magnetic, mechanical, and other properties of nanomaterials differ from those of the corresponding bulk materials (Figure 1a). Because of the unique properties of magnetic nanoparticles, such as superparamagnetism, high saturation magnetization, and high effective surface area, they are mainly used as contrast agents, to improve image contrast, and as carriers for drug delivery in disease treatment. In addition, when injected into the body, magnetic nanoparticles can generate heat energy through the use of an applied magnetic field to kill cancer cells, avoiding the damage to normal cells observed in conventional chemotherapy and inhibiting cancer cell growth by MFH. Because of their excellent magnetic properties, the application of nanomaterials is constantly being improved and refined [15,16,17]. All substances have a certain degree of magnetization, which is usually dependent on the material’s atomic structure and surrounding temperature, and the magnetic susceptibility (χ) can be used to express the difficulty of magnetization. When a material is placed under an applied magnetic field (H), its magnetization (M) will change, and the relationship between the two is as follows (Figure 1b):(1)M=χH

A magnetic material has a “magnetic domain”, in which the crystalline structure of the energy state itself is divided into several different regions, and these magnetic domains are all oriented in the same direction. Nevertheless, the order of each magnetic domain is not necessarily the same, and there will be mutual offset, as shown in Figure 1c. Assuming that the net magnetic moment is precisely zero, the material is not magnetic. Conversely, when the net magnetic moment is not zero, the material is magnetic [18]. Most materials have the property of being weakly magnetic, even in the absence of an applied magnetic field. The former has an approximate magnetization rate ranging from 10^−6^ to 10^−1^ in order of magnitude, while the latter only ranges from 10^−6^ to 10^−3^ in order of magnitude. In contrast, some materials can exhibit highly magnetic properties under the action of weak magnetic fields, or even without the application of magnetic fields, such as in the case of ferromagnetic, ferrimagnetic, and antiferromagnetic materials (Figure 2a). In such cases, only a minimal magnetic field is needed to saturate magnetization, and the representative materials are iron, cobalt, and nickel.

From a microscopic point of view, a large-size ferromagnetic material with multiple magnetic regions exhibits a minor hysteresis effect, as shown in Figure 2b. When the size of the material is reduced to a single domain (generally nanoscale; the critical size varies from material to fabric), the hysteresis effect is the largest (i.e., it has the most substantial coercive force). However, when the scope continues to shrink, the coercive force decreases to zero (i.e., no hysteresis occurs). With the change in the external magnetic field, the data points obtained from the magnetization process of the superparamagnetic nanoparticles can form a hysteresis curve, as shown in Figure 2d. Therefore, the superparamagnetic nanoparticles are not magnetic at room temperature without the applied magnetic field, and when the applied magnetic field is removed, the material’s magnetic properties immediately disappear. For example, in iron oxide, nanoparticles usually need to reach a size of several nanometers to become superparamagnetic, as shown in Figure 2c [19].

### 2.1. Properties of Iron Oxide Nanoparticles

Iron oxide nanoparticles are widely known examples of nanomaterials. Iron tetroxide (Fe_3_O_4_) is a biocompatible material that has been known of and used in biomedicine for almost 40 years, and it is approved for use in the human body based on its safety profile. Fe_3_O_4_ magnetic nanoparticles are water-soluble and can enter delicate tissues; they are commonly produced by coprecipitation (aqueous phase), thermal decomposition (organic phase), and synthetic methods (Figure 3a–c) [11]. For use in biomedical applications, nanoparticles must be: (1) non-biotoxic; (2) water-soluble; and (3) biocompatible. The biomedical applications in which nanoparticles of iron tetroxide are applied are diverse [20]. The most remarkable instances are those where their magnetic properties are exploited, for example, in thermo-therapy and drug magnetic guidance therapy. For example, in the case of a rat with a tumor on its back, we can inject magnetic nanoparticles through the tail and use carefully positioned magnets around the rat to achieve guided drug treatment for the cancer [21,22,23].

Heat therapy aims to increase the temperature to a level that is tolerated by normal human cells but not cancer cells, at which point the cancer cells begin to die. The current research indicates that thermotherapy can effectively eliminate tumors that are smaller than 7 mm [24]. Still, in clinical trials, uniform tumor heating is impossible on animals with larger tumors (15 mm). The cancer is unevenly heated because of the uneven shape of the tumor, and it is hard to effectively destroy the entire cancer all at one time during treatment, which results in continued tumor growth [25]. Because of this, we want to develop magnetic particles that can be used to accelerate and increase the temperature of the cancerous tissue. One possibility is to dope magnetic particles into ceramic nanostructures to generate nanocomposite materials. The composite particles enable the faster elimination of cancer cells, thus achieving more effective treatment, and they are expected to more efficiently inhibit the formation of tumors. In addition, we can use the method to encapsulate magnetic nanoparticles and drugs at the same time. This problem can be resolved through ceramic-material-compounding technology, which is thermosensitive and biocompatible. Moreover, microcellular surface carriers with specific surface modifications can deliver magnetic nanoparticles and drugs to specific tumor cells. The magnetic nanoparticles are heated to 40 °C for a few seconds under an applied magnetic field, upon which the ceramic material releases the drug and magnetic nanoparticles into tumor cells [21]. Therefore, this two-pronged approach allows incorporating an additional aspect to drug therapy.

### 2.2. Properties of Iron–Platinum Nanoparticles

Iron–platinum nanoparticles are a magnetic material that is used in recording media with the chemical formula FePt. There are four phases of ferroplatinum: the (1) unordered γ-phase; (2) ordered paramagnetic γ1-Fe_3_Pt phase (L12); (3) ferromagnetic γ2-FePt phase (L10); and (4) antiferromagnetism γ3-FePt_3_ phase (L12). The structure of the FePt_3_ phase (L12) depends on the FePt atomic ratio. The structure of unordered Fe platinum is chemically disordered face-centered cubic (FCC), and that of ordered Fe platinum is chemically ordered face-centered tetragonal (FCT) [26], as shown in Figure 4 [27]. In addition, the atomic lattice position of the unordered face-centered cubic is determined by the percentage of Fe and Pt atoms that form a soft magnetic structure with a small coercivity field [28]. In contrast, in the ordered face-centered cubic in iron–platinum, the iron atoms are stacked at positions (0, 1/2, 1/2) and (1/2, 0, 1/2), the platinum atoms are stacked at positions (0, 0, 0) and (1/2, 1/2, 0), and the atomic radii cause the lattice to expand in the a-axis and compress in the c-axis [29,30,31,32]. The magneto-crystal anisotropy coefficient (Ku) can reach 107 Jm^−3^ [33], which is the highest among the existing hard magnetic materials, due to spin–orbit coupling and hybridization interactions between the 3D orbital domain of Fe and 5D orbital domain of Pt [34,35]. This alignment provides FePt with higher chemical stability than Fe, Co, or other materials (e.g., Fe_3_O_4_) with high coercivity fields [36]. The high Ku of FePt can allow superparamagnetic phenomena to be avoided when the particle size decreases [37]. In addition to having superior superparamagnetic properties, FePt nanoparticles also have high absorption coefficients for X-rays (Pt absorption coefficient at 50 keV: 6.95 cm^2^/g). Chou et al. injected 12 nm FePt nanoparticles into mice with tumors by tail-intravenous injection, and the contrast between the MRI and computed tomography (CT) images was substantially improved, which indicates that we can use FePt to track the location of the material in two diagnostic MRI and CT imaging modalities to detect the area in which the MFH and drug release are taking place.

## 3. Surface Modification of Iron-Based Nanoparticles

The key to the technology is how to use ligands for surface modification and increase the function of magnetic nanoparticles. Generally, two methods are used: (1) Crosslinkers or spacer molecules as well as polymer ligands are used to form covalent connections [38]. The body is modified on the surface of the magnetic nanoparticles to include iron nanoparticles as the core and ligands as the shell [39]. The affinity between nanoparticles and polymer ligands depends on the type and quantity of the ligands on the surfaces of the nanoparticles; thus, how to make and select the surface ligands for linking is important [40]. Amines, carboxylates, hydroxyl groups, and thiol groups are commonly used as ligands [41]. In some cases, additional spacer molecules or crosslinking agents are required to facilitate bonding of the nanocomposites [42]. (2) In layer-by-layer coating [43], with magnetic nanoparticles as the core, other materials are coated, layer by layer, around the nanoparticles based on the electrostatic attraction between opposing charges [44]. The advantages include the ability to fabricate a single-layer structure and adjust the thickness of the functional shell [45]. According to the above two approaches, we take the carboxylation of chitosan to covalently bond to the surfaces and core–shell structures of the nanoparticles as an example, and we discuss the advantages of ceramic materials for modification of iron-based nanoparticles in the next section (Figure 5).

Chitosans are natural polysaccharides with hydrophilic, biocompatible, biodegradable, and antibacterial properties. They have a good affinity for many biomolecules, which makes them suitable for various biomedical and biotechnology applications. Degradable polymers are more commonly used for the controlled release of drugs [46]. Polysaccharides are nontoxic and biodegradable natural polymers that form particles to coat drugs in acidic environments, such as in the stomach, where they act as antacids to prevent acid damage to drugs. Therefore, they are an ideal material for drug-release-control systems. For drug-targeting applications, magnetic nanoparticles modified with chitosans can adsorb the anticancer drug epirubicin, which indicates a strong interaction between chitosans and epirubicin. In epirubicin-adsorption experiments, the equilibration time is only a few minutes, which means that there is no intrapore diffusion resistance in the adsorption process. Through regulation of the acidic environment in cancer cells at a pH of 4, chitosan is subjected to disintegration. Epirubicin adsorbed on nanomagnetic carriers is expected to be released in in vivo experiments to achieve therapeutic cancer effects [47].

The surface modification of iron-based nanoparticles with core–shell structures is close to that of iron-based nanocomposite particles combined with ceramic materials, which is the focus of this review: i.e., enhancement of the applicability of iron nanoparticles by incorporating other materials. Here, we search for an example of self-assembled nanocomposite materials for iron core–gold shells to link the advantages of ceramic materials combined with iron nanoparticles [48]. The iron core–gold shell composite nanoparticles are selectively toxic to cancer cells. Still, after being placed in water or air for a suitable period, they are no longer harmful to cancer cells [49]. Researchers found that freshly produced iron core–gold shell composite nanoparticles are not toxic to cancer cells when placed in water. Water molecules will penetrate through the grain interface of the gold shell and react with iron at the gold–iron interface to produce ferrous ions, which are gradually released to kill cancer cells. However, the dissolved oxygen in the water will also spread to the gold–iron interface through the grain interface of the gold shell, oxidizing the iron into iron oxide, which forms a protective layer to prevent the continued production and release of ferrous ions and, thus, no longer having a toxic killing function and achieving the effect of self-liquidation [50]. Likewise, protection is also provided by the ceramic material compounded with iron-based nanoparticles. In addition, due to its porous nature, the ceramic material can also offer drug loading and delivery of iron nanoparticles, similarly to chitosan mentioned above.

## 4. Combination of Iron-Based Nanocomposite Particles with Ceramic Materials

Clay minerals are one of the most important industrial minerals in nature. In this section, we focus on white clay ore-containing water, which is mainly made from aluminosilicate minerals, such as feldspar, and is formed by climate or water heat capacity. Because clay is easily shaped in moist conditions and can be cured after sintering, many products, such as road bricks and sewage pipes, contain clay minerals as raw material. In addition, clay minerals are white and resistant to high temperatures; thus, they are used in the porcelain, paper making, rubber, and refractory industries. As a new type of drug delivery system, ceramic nanocarriers have high mechanical strength, good body response, and low or non-existing biodegradability. Ceramic nanocarriers can protect the drug and the composite nanoparticles from pH and temperature effects. However, despite the high biocompatibility shown in current studies, there is still a lack of information on their clinical use [51]. The research journey for future applications of ceramic nanocarriers is still long; thus, this section will focus on the improvements brought by ceramic materials composites.

### 4.1. Bioactive Glasses

The development of suitable biomaterials for application in bone regeneration and disease treatment is a substantial challenge in current regenerative medicine. Synthetic biomaterials can be prepared using flexible synthetic methods to combine the best possible properties, such as bioactivity, degradation, and controllable drug delivery [52]. This allows various imaging, cell-specific-targeting, and controlled-drug-release functions to be incorporated into a single platform designed for simultaneous tracing and convenient therapeutic use without losing the individual properties of each component [53,54]. However, combining these different functions on the same platform is extremely difficult, which is because competition between the various functional groups could be generated when on the same material platform. As the application of a synthetic biological scaffold, bioactive glasses (BGs) are the leading group of surface-reactive glass–ceramic biomaterials. Due to the excellent biocompatibility of these glasses, they have been widely investigated by researchers for use as implant materials in the human body to fill and repair bone defects [55]. BGs were discovered in 1971 by the research group of Hench [56]. In the physiological environment of the human body, BGs can react with simulated body fluid (SBF) to form dense biologically active hydroxyapatite (HA) layers on their surface and biologically bond with damaged bone. HA is the main mineral component of bone that leads to effective physical interactions and fixes bone tissue onto the material surface [57]. Researchers have developed different families of BGs for bone tissue restoration and replacement because such materials do not cause biological toxicity, inflammation, or elicit an immune response [58]. Because of these characteristics, BGs have been extended to many different applications in the medical field, such as implants in theoretical bone repair, tissue engineering, drug delivery, and bone cancer treatment [52,53].

In 2004, Yan et al. used advanced science and proficient technology to develop a novel family of biomaterials called mesoporous bioactive glasses (MBGs). Compared with conventional BGs, MBGs have higher specific surface areas and pore volumes. MBGs exhibit improved bioactive behavior with even faster apatite phase formation than conventional BGs [55,56,59,60,61,62,63]. In 2006, Chang et al. produced a well-ordered MBG as a drug delivery carrier [64]. In numerous recently published studies, researchers have developed MBGs as a biomaterial extensively applied in drug delivery systems and bone tissue engineering [53,65,66,67,68,69,70]. In the latest technique, Zhang et al. fabricated a composite scaffold containing mesoporous bioactive glass to encapsulate magnetic Fe_3_O_4_ nanoparticles by 3D-printing technology. According to the results, the MBG scaffold structure comprises uniformly sized 400 μm macropores. The magnetic Fe_3_O_4_ nanoparticles can be incorporated into the scaffold without affecting its hydroxyapatite mineralization ability while endowing it with excellent magnetic heating ability. In addition, the pore structure can be loaded with doxorubicin (DOX), which is an anticancer drug, and it can thus be used for local drug delivery therapy. The 3D-printed Fe_3_O_4_/MBG scaffold shows potential versatility for enhanced osteogenic activity, local anticancer drug delivery, and magnetothermal therapy [71].

### 4.2. Biocompatible Nanolayer Ceramics

The basic structural layer of nanolayer ceramics, which is composed of silicate minerals, consists of a silicon–oxygen tetrahedron and an aluminum–oxygen octahedron, each of which has three oxygen atoms in the same plane and one oxygen atom at the top. The aluminum–oxygen octahedron consists of a stack of oxygen atoms and hydroxide ions, with the cation at the center of the octahedron and each cation bonded to six oxygen atoms (or hydroxide ions) to form an octahedron, as shown in Figure 6a [52]. According to the ratio of the tetrahedral and octahedral sheets contained in each layer of the clay minerals, they can be divided into two types: (1) the 1:1-layer type, in which the interlayer formed by stacking one tetrahedral sheet and one octahedral sheet is called the TO layer, whereas the tetrahedral plane on top and adjacent octahedral OH surface below form the coordination of OHO, which is the simplest crystalline structure of layered silicate clay minerals [72]; (2) the 2:1-layer type, in which each layer is composed of two tetrahedral sheets that are sandwiched between octahedral sheets, which forms a three-layer structure of TOT, similar to a sandwich [73].

For example, montmorillonite is a 2:1-layer silicate, and each molecular formula has from 0.2 to 0.6 units of charge. The interlayer cations of montmorillonite, such as Na^+^, Ca^2+^, and Mg^2+^, are exchangeable cations with high hydration. The interlayer distance is about 9.6 × 10^−1^ nm when there are no water or polar molecules in the interlayer, whereas the interlayer distance of montmorillonite containing divalent cations (Ca^2+^ or Mg^2+^) increases to 14 × 10^−1^ nm at an average humidity of 40–60% because the interlayer contains two water molecules in the water layer. If monovalent cations (Na^+^) are present in the interlayer of montmorillonite under the same humidity conditions, then the interlayer distance is 12.5 × 10^−1^ nm [74]. Another characteristic of montmorillonite is that it contains many exchangeable cations. Because the negative charges in the montmorillonite structure are concentrated in the central octahedral layer, the interlayer cations are weakly bound and can be easily replaced. The typical chemical formula of montmorillonite is (1/2Ca,Na)_0.7_(Al,Mg,Fe)_4_(Si,Al)_8_O_20_(OH)_4_.nH_2_O, in which Ca^2+^ and Na^+^ are exchangeable cations. The theoretical chemical composition is 49.0% SiO_2_, 23% Al_2_O_3_, and 0.3% Fe_2_O_3_.

For ceramic kaolinite, the basic structural layer is composed of silicate minerals, each of which has three oxygen atoms in the same plane and one oxygen atom at the top. The aluminum–oxygen octahedron consists of a stack of oxygen atoms and hydroxide ions, with the cation at the center of the octahedron and each cation bonded to six oxygen atoms (or hydroxide ions) to form an octahedron [29]. We present the structure of kaolinite in Figure 6b, which consists of a layer of silica–oxygen tetrahedra and a layer of alumina–oxygen octahedra that are connected by a standard oxygen linkage to form a bilayer structure. In contrast, the layers are covalently bonded by providing oxygen atoms on the silica–oxygen side and hydroxide ions on the alumina–oxygen side to form hydrogen bonds. As previously mentioned, kaolinite has a stable chemical structure, uniformly distributed pore structure, and high adsorption capacity, and it can adsorb different substances in its layered structure, such as FePt nanoparticles or chemotherapeutic drugs. With the adsorption effect provided by kaolinite, FePt nanoparticles can be highly concentrated in a specific space. The nanoparticles can effectively accumulate in a magnetic-field environment according to the influence of the magnetic force to achieve a magnetically controlled MRI effect. If magnetic control is used to guide the accumulation of drugs into tumor tissue, then the side effects caused by chemotherapeutic drugs can be substantially reduced. Currently, a single treatment is not enough to achieve the substantial inhibition of tumor tissue. Cocktail-style therapies have become standard in current cancer treatments. FePt nanoparticles have excellent magnetocaloric effects, and when combined with kaolinite, their heating capacity can be substantially increased. The temperature can be increased to nearly 50 °C; thus, hepatocellular cancer cells can be killed using MFH. At the same time, if kaolinite is loaded with chemotherapeutic drugs, such as Dox, then the system can further inhibit cell growth at the tumor center.

### 4.3. Biocompatible Nanotube Ceramics

Hardystonite or akermanite nanotubes belong to the kaolinite group of aluminosilicate clay minerals and were discovered by the Belgian geologist d’Omalius d’Halloy and named by Pierre Berthier in 1826. Depending on the mining site and geological conditions, they can be tubular, spherical, or plate-like particles. Among these forms, the most representative is the tubular form with cavities, which has received substantial attention in various research fields due to its particular morphology, ease of mixing with multiple polymers, and good biocompatibility. The basic structure of the silicate mineral composition consists of a silicon–oxygen tetrahedron and an aluminum–oxygen octahedron, each of which has three oxygen atoms in the same plane and one oxygen atom at the top.

The cation is located at the center of the octahedron, and each cation forms a bond with six oxygen atoms (or hydroxide ions) to form the octahedron, as shown in Figure 6c [27]. Hardystonite is similar to kaolinite, but the layers in hardystonite are separated by a single layer of water molecules and are classified according to their hydrated state. Akermanite is a hydrate elite (10 × 10^−1^ nm), and when dried, it irreversibly loses the interlayered water to form a dehydrated elite (7 × 10^−1^ nm), which is more stable than the hydrated akermanite. The structure is caused by the mismatch between the silica–oxygen tetrahedra and aluminum–oxygen octahedral sheets in the layers. The tetrahedra and octahedra are connected through sharing of the top oxygen of the tetrahedra. This stress is transferred to the Si plane and the base oxygen plane through the Si–O bond, but it is also reduced by the angular elasticity of the Si–O bond. In most of the current tubular materials that are compounded with iron-based nanoparticles, carbon nanotubes are used as the carrier, and the hardystonite or akermanite structure is coated with a layer to multiply the nanoparticles. Alternatively, the iron-based nanoparticles can be doped with alpha-Al_2_O_3_ crystal to affect carbon nanotube growth in polycrystalline ceramics. Celik et al. prepared Fe-doped Al_2_O_3_ ceramics of different textures through templated grain growth and synthesized them into carbon nanotubes via catalytic chemical vapor deposition [75]. According to the experimental results, this novel nanocomposite material has the potential to be used for future biomedical diagnostic and drug delivery applications.

## 5. Magnetic Resonance Imaging (MRI) with Ceramic Material Composite Iron Nanoparticles

Among the many screening and diagnostic methods, magnetic resonance imaging (MRI) can provide high-resolution images of the liver without the need for ionizing radiation. Consequently, MRI is the best choice for initial tumor diagnosis. For this reason, iron-based nanomaterials have become candidates for MRI imaging because of their excellent T2 contrast ability [76]. However, most of the commonly used magnetic vibrating carriers currently available in the market are iron oxide particles or strontium ion complexes, which may cause side effects, such as nausea, allergic reactions, and kidney injury [77]. To address this, the goal of current research is to improve the performance of magnetic carriers through the development of a multifunctional composite nanocarrier that can be used for high-resolution MRI with low toxicity and a therapeutic effect [78].

Positively charged atomic nuclei spin in random environments. In this case, the nuclear spin axes are arranged in a random pattern, and when placed in a static magnetic field, the nucleus spins in the direction of the applied magnetic field. The spin frequency is called the Larmor frequency, and it is related to the properties of the nucleus itself and is proportional to the strength of the applied magnetic field. The effect is that the iron-based nanoparticles are subjected to single-axial compressive stress, and the mechanical stress causes the rearrangement of the magnetic dipoles of the iron-based nanoparticles and places them parallel with the direction of the ceramic layer. Therefore, in human MRI, the degree of saturation magnetization along the magnetization direction depends on the hydrogen atoms, which are the main source of nuclei because the human body is mainly composed of water, and it is the hydrogen atoms in water that help in visualizing the image. As the magnetization vector of the atomic nucleus gradually increases and then stabilizes during the spinning process, the nucleus will resonate if disturbed by a fixed-frequency radio frequency (Figure 7). The resonance effect, which is limited by the ceramic material, brings higher saturation magnetization to the iron-based nanoparticles, thereby enhancing their ability for T2-weighted MRI diagnostic imaging.

## 6. Magnetic Fluid Hyperthermia (MFH) with Ceramic Material Composite Iron Nanoparticles

Thermal treatments include laser treatment, focused ultrasound treatment, microwave treatment, and use of radiofrequency probes. These treatments aim to raise the temperature of the tumor to 43–46 °C to achieve the effect of thermal therapy [79,80,81].

However, the above-mentioned thermal therapy systems are macroscopic heating systems, which generally have the disadvantages of low thermal efficiency and being easily limited by the tumor volume, which result in uneven heat-field distribution [82]. In addition, some new methods have been developed, including pulsed laser, infrared, and magnetic-field-guided heat therapies [83]. Examples of heat sources include metal nanoshells, nanorods, and carbon nanotubes. In short, when these energy-absorbing materials reach the tumor, they can be irradiated from outside the body using strong energy sources (e.g., near-infrared laser). When the materials absorb this energy, it is converted into heat energy to increase the temperature of the tumor surface and destroy the tumor structure, thus achieving the effect of heat therapy [84]. However, this treatment method can only treat tumors close to the body surface and is inadequate for deeper tumors.

In recent years, several independent research groups have been developing a method termed magnetic fluid hyperthermia (MFH), which involves use of a magnetic fluid together with an alternating magnetic field to treat tumors [85]. This method improves upon the drawbacks mentioned above, killing cancer cells without affecting the adjacent normal tissues [86,87].

### 6.1. Principles of MFH

Magnetic nanoparticles exposed to alternating current (AC) magnetic fields generate heat by hysteresis loss [88,89]. However, not all magnetic nanoparticles generate heat in this way. For magnetic nanoparticles with multiple magnetic domains (e.g., 2- and 3-valent iron), the heat in an AC-field environment is generated through hysteresis loss, and for magnetic nanoparticles with single magnetic domains (e.g., single-domain ferric tetroxide nanoparticles), the heat is generated through Néel relaxation and Brownian relaxation. The reason heat is not generated by hysteresis loss for magnetic particles is that because of their superparamagnetic properties, they have a single magnetic domain and fixed magnetic moment direction [90]. 

Therefore, the heating principle can be divided into hysteresis loss, Néel relaxation, and Brownian relaxation [91]:(A)Hysteresis loss: When a material has multiple magnetic domains, the direction of the magnetic moment becomes singular and the same as the magnetic field when an AC magnetic field is applied. When the magnetic-field strength changes, the resulting hysteresis curves do not overlap, which results in heat release;(B)Néel relaxation: When a material is a single-domain superparamagnetic material, the inner nucleus rotates and overcomes the energy barrier E = KV when an AC magnetic field is applied, where K is the anisotropy constant, and V is the volume of the particle. The thermal energy is released when it returns to the original magnetic moment direction;(C)Brownian relaxation occurs in materials with multiple or single magnetic domains when an applied magnetic field is applied, which causes the particles to rotate and rub against the external medium and release thermal energy. Therefore, the characteristics of Brownian relaxation are related to the solution viscosity, as shown in Figure 8a.

The following equation is the heat loss (P) for a single-magnetic-domain material, as shown in Equation (2).
(2)$$P=\frac{V{\left(M_{s}H\omega \tau \right)}^{2}}{2{\tau k}_{b}T\left(1+\omega^{2}\tau^{2}\right)}$$
where V is the particle volume; M_S_ is the saturation magnetization value; H is the AC-electromagnetic-field strength; ω is the angular frequency of AC; τ is the relaxation time; and k_b_ is the Boltzmann constant. When ωτ = 1, we obtain the maximum heat loss value. When the saturation magnetization is more substantial, the heat loss is more extensive:(3)$$\frac{1}{\tau }=\frac{1}{\tau_{B}\;}+\frac{1}{\tau_{N}};\;\tau =\frac{3\eta \mathrm{VH}}{\mathrm{kT}};\tau_{N}=\tau_{0}\exp\left(\frac{\mathrm{KV}}{K_{b}T}\right)\;$$
where η is the medium viscosity; VH is the particle hydration volume; k_b_ is the Boltzmann constant; V is the particle volume; K is the anisotropy constant; T is the absolute temperature; and τ_0_ is the time constant. We use the specific adsorption rate (SAR) to obtain the thermal energy generated by the predicted material. The efficiency of conversion from energy to thermal energy of the magnetic nanoparticles in an AC-magnetic-field environment is related to the AC-magnetic-field frequency, particle size and surface modification [91].

### 6.2. Treatment with MFH

When the local tumor temperature is increased to 41–46 °C by the magnetofluid in an AC-magnetic-field environment, the reason for using heat to treat tumor cells can be easily understood, as shown in Figure 8b,c below. One of the differences between tumor cells and normal tissues is that tumor cells receive more nutrients through continuous neovascularization [92]. However, most of these new blood vessels are disorganized and functionally abnormal; therefore, when heat is applied, tumor cancer cells, similarly to normal cells, increase their blood flow by vasodilatation to carry away the heat; however, this process is inefficient, and heat is retained in the tumor tissue. Therefore, as the temperature increases, neovascularization in the tumor tissue is continuously disrupted, which results in reduced blood flow and heat retention. Finally, the tumor cannot obtain nutrients to achieve the therapeutic effect. As shown in Figure 8e, after heat treatment in rats, the blood flow in the tumor cells is reduced, while the blood flow in normal cells is increased by 8–10 times. Therefore, poor heat dissipation ability may cause heat to become trapped inside the tumor, increasing the temperature, which further affects the pH, pO_2_, and nutrient supply and leads to cell death [93].

## 7. Drug Delivery with Ceramic Material Composite Iron Nanoparticles

To reach the desired treatment site in clinical chemotherapy, high doses of drugs are usually required, increasing the risk of nonspecific toxic reactions and other physiological side effects that cause additional pain to patients. Therefore, how to efficiently deliver low-dose medications to the desired treatment site has long been a research direction for pharmaceutical companies and laboratories. The magnetic-drug-targeting technique, which uses magnetic nanoparticles in conjunction with an external magnetic field, has recently gained attention (Table 1). Magnetic nanoparticles are mainly used as drug carriers in this application. In general, magnetic nanoparticles containing drugs or antibodies are intravenously injected into the body, transported through the circulatory system, and finally concentrated at the site of the applied magnetic field, as shown in Figure 8d. In this way, more drugs can be directly focused on the lesion and then released through the drug-release mechanism. Lubbe published a study in 1996 in which they performed the first human clinical trial using magnetic drug targeting [94]. They used an intravenous infusion of magnetic particles (100 nm particle size; starch) immobilized with epirubicin (a tumor treatment drug) in solution and placed a permanent magnet with a magnetic flux density of 0.8 T close to the treatment site. According to the results, they could successfully guide the magnetic particles to the target area in more than half of the subjects. FeRx Inc. has attempted to commercialize this technology for cancer treatment, and it is currently undergoing clinical trials [95]. Biological applications of ceramic material composite iron nanoparticles depend on the carrier’s biological toxicity and the composition of the ceramic structure, both of which necessitate the use of biocompatible materials. The main categories of application are biosensors, oxidative stress cytoprotection, bacterial disinfection, and cancer treatment.

### 7.1. Cancer Therapy

#### 7.1.1. Bone Cancer

Currently, bone cancer is primarily treated by shaving the affected area. This approach is likely to cause disease recurrence because the cancer cells are not entirely eradicated. Therefore, the treatment process is usually combined with chemotherapy. However, most anticancer drugs have low solubility and severe drug side effects. These disadvantages have motivated the design of a controlled drug delivery system, as shown in Figure 9a [110]. The purpose in the research of Farzin et al. was to enhance the treatment impact. In their study, the researchers used iron oxide nanoparticles as a mechanism for controlled drug release, which can be combined with BGs to exploit the magnetic field to control the distribution of the anticancer drug in the bone tissue and prevent its release in other undesired locations [111]. To elaborate the use of BGs as a drug platform, various modified methods may be helpful in multiple medical applications, such as implantation in medical bone restoration, tracking postoperative surgery, and treating bone cancer [112]. To increase the accumulation of iron-based materials at the tumor site, researchers have developed modifications of the specific targeting molecules in ceramic materials. In native bone cancer, the folate receptor is not overexpressing; however, most bone tumors are metastatic. Therefore, the material with grafted folate molecules can still be used to treat bone cancer transferred from other cancers [113]. The MG63 (human osteosarcoma) cell line is supposedly a suitable in vitro test model for bone cancer. However, MG63 lacks the folate receptor; no noticeable difference in the endocytosis process was observed when folate molecules were grafted onto our material. This distinction between normal and cancer cells has made FA an attractive ligand for specific targeted bone cancer drug delivery [114].

#### 7.1.2. Liver Cancer

Hepatocellular carcinoma (HCC) is a primary malignant tumor of the liver cells. According to the National Cancer Institute SEER database, the average five-year survival rate for patients with HCC is 19.6%, and the survival rate for advanced metastases is as low as 2.5%. Following early diagnosis, treatment can be provided through local-area therapy, including surgical resection, radiofrequency ablation, transvenous chemoembolization, and liver transplantation. HCC is usually diagnosed at an advanced stage, when the tumor cannot be removed, which renders these treatments ineffective. Liver cancer is the fifth most common cancer and the fourth leading cause of cancer-related deaths worldwide [115]. There are two major types of primary liver cancer, HCC and intrahepatic cholangiocarcinoma (ICC), and less common cancers such as angiosarcoma, hemangiosarcoma, and hepatoblastoma. HCC accounts for more than 80% of primary liver cancer cases worldwide, and secondary liver cancer occurs when tumors from other parts of the body metastasize to the liver. Although breast, esophageal, stomach, pancreatic, lung, kidney, and several other cancers can metastasize to the liver, most secondary liver cancers originate from colorectal cancer. Approximately 70% of patients with colorectal cancer will develop secondary liver cancer [116].

HCC is the most common type of chronic liver cancer in adults and the most common cause of death in patients with cirrhosis. Unlike other organs or tissues in the human body, the nerves of the liver are distributed on the surface, with few inside the liver. Therefore, when a small tumor grows in the liver, it is almost painless and does not show any symptoms. Without regular checkups, it is easy to overlook the potential threat of the tumor tissue [117,118]. Among the various screening and diagnostic methods, MRI can provide high-resolution liver images without ionizing radiation. HCC exhibits a high-intensity pattern in T2-weighted images. Owing to the selective role of the hepatobiliary system, the application of iron-based nanoparticles leads to an increased accumulation of iron in the liver, thereby increasing the sensitivity of MRI for liver imaging. Chan et al. have developed a ceramic material compounded with FePt nanoparticles, which is a superparamagnetic iron-based nanomaterial contrast agent that is suitable for HCC diagnosis. With superparamagnetic, low-toxicity, biocompatible, and adaptable characteristics, iron-based nanocarriers are widely used in biomedical applications, such as imaging, differentiation, fluorescent labeling, clinical diagnosis, and drug delivery [105]. To enhance and optimize the application of FePt nanoparticles in MRI, a magnetic kaolinite and montmorillonite composite material is used to adsorb a large amount of FePt nanoparticles in realizing optimal MRI conditions (Figure 10a,c). The fine particles of kaolinite and montmorillonite have stable chemical structures, uniformly distributed pore structures, and high adsorption capacities. They can adsorb different substances in their layered structures, such as FePt nanoparticles or chemotherapeutic drugs. The novel ceramic-combined FePt nanocomposites exhibit enhanced magnetic flux, as seen in Figure 10b (according to the vibrating sample magnetometer, the magnetic field of the nanocomposites is approximately 78% higher than that of the FePt particles). FePt nanoparticles have excellent magnetocaloric effects. Their heating capacity can be substantially increased when combined with kaolinite and montmorillonite. The temperature can be increased to nearly 50 °C, and thus, HCC cancer cells can be killed by MFH [108]. In addition to MFH, HCC can also be diagnosed in mice using high-precision MRI after enhancement of the magnetic flux through composite materials (Figure 10d). 

#### 7.1.3. Breast Cancer

Medical research has led to the development of powerful treatments against breast cancer. Thanks to advances in science, we can pinpoint the specific weapons that are most effective for individual patients, which is a process that is referred to as precision medicine. Precision medicine is a growing trend in modern medicine, and it involves the creation of treatment plans that are best suited to each individual’s disease, environment, and lifestyle. Selecting treatments for individual patients and developing care plans that are tailored to individual needs is not a new concept. The most dramatic changes have come from our knowledge of genetics and cancer biology, including that of breast cancer. Precision medicine is already being used in breast cancer treatment. For example, iron-based nanoparticles can detect whether a breast cancer tumor cell is making excessive amounts of HER2 protein. If it is, then the tumor is classified as HER2-positive, which can be effective if a drug-targeting HER2 is used.

Another example is genetic testing for women with a strong family history of breast cancer. Specific genetic mutations, such as *BRCA1* or *BRCA2*, can substantially increase the risk of developing breast cancer. Women with these genes can reduce the cancer risk by initiating preventive measures more often and earlier, such as through mammography and MRI using iron-based contrast for the observation of T2-weighted MRI images. In addition, we can use these modern genetic tests for breast cancer to determine the breast-cancer-recurrence rate and whether post-surgical chemotherapy would be beneficial. Increasingly, such tests are guiding physicians in making treatment decisions. As more biomarkers are identified and more treatments are developed, the precision of breast cancer care will become more accurate. When these biomarkers are used to guide iron-based materials into breast cancer cells, the iron nanoparticles can be used to thermally kill the tumor tissue by MFH to obtain targeted therapeutic results [119]. Wang et al. developed a heat-shrinkable, injectable biodegradable material composed of hydroxypropyl methylcellulose (HPMC), polyvinyl alcohol (PVA), and Fe_3_O_4_. The authors chose MB-231 for the in vitro experiments to show that the ablation of tumors is positively correlated with the weight of the HPMC/Fe_3_O_4_, iron content, and heating time. This novel, safe, and biodegradable material will facilitate the technological transformation of MFH, and it is also expected to introduce new concepts to the field of biomaterial research. Moreover, Tseng et al. used hydroxyapatite (HAP) as a drug carrier for breast cancer treatment via MFH and chemotherapy. The authors developed bifunctional nanoparticles (Pt–Fe-HAP) made of HAP containing iron and platinum ions for combination therapy [120].

### 7.2. Promotion of Osteoblast, Fibroblast, and Bone Marrow Mesenchymal Stem Cell Proliferation

Biocompatible synthetic bone grafting based on BGs is widely used in orthopedics and dentistry. Clinically, similar results to those shown in Figure 9b,f, can be achieved using BGs alone or in combination with other bone grafts for filling bone defects in periodontal surgery with transformation into HA via the body fluid immersion method [121]. A common ceramic material that contains calcium phosphate, HA is biocompatible and bone resorptive; hence, it is the bone substitute that is most widely used in bone tissue engineering, functioning as a platform facilitating bone regeneration [122,123,124].

The technological development of multifunctional materials has been the focus of the research in recent years. For example, silica materials embedded in a light-induced fever agent can be exploited in both photothermal therapy and near-infrared fluorescence imaging [125]. As another example, Fe_3_O_4_ nanoparticles are of use in thermal drug release and MRI when combined with a temperature-sensitive polymer and when folic acid (FA) molecules are grafted onto their surface [126]. HA nanoparticles are suitable for T1-weighted MRI when processed to contain europium (Eu^3+^) and gadolinium (Gd^3+^) ions, and they can also be modified with FA molecules to target cancer cells [127]. Moreover, researchers have also reported the use of multifunctional HA nanorods in actual practice [128]. The potential for great demand is beyond doubt, given the scarcity of ceramic substrates combined with iron-based nanoparticles. Wang et al. generated BGs using 3D printing after loading iron-based nanoparticles onto the surface, and they observed bone tissue repair and regeneration through MRI (Figure 9c). Through microscopy, they maintained the proliferation and growth state of the bone marrow mesenchymal stem cells by creating muffin-like 3D block structures and loading them with iron nanomaterials (Figure 9d,e). They confirmed that the ceramic material could be used to repaired the bone defect by generating HA under biomimetic body fluid.

### 7.3. Other Biological Applications Related to Drug Release

Owing to the sensitivity and detection limits, we cannot use conventional biochemical assays to distinguish slight differences. Zuo et al. produced monoatomic iron test strips by the dropwise addition of an aqueous hydrogen peroxide solution to detect the catalytic effect of butyrylcholinesterase in combination with image capture using a cellphone [129]. The combination of the cell phone-photo function to obtain fluorescent images is expected to promote further development in the field of portable detectors. Ma et al. synthesized four nitrogen–ligand monoatomic iron–carbon materials to mimic hydrogen peroxidase and superoxide dismutase to break down intracellular reactive oxygen species and prevent apoptosis [91]. Xu et al. used an organometallic framework containing monatomic iron to suppress the inflammatory response and accelerate tissue growth in a wound-healing experiment [130]. 

## 8. Conclusions

Of the substantial applications of magnetic nanoparticles combined with ceramic materials in the biomedical field that were introduced in this review, most are still in the phase of clinical testing or at the laboratory stage, except for contrast agents, which are currently available in commercial form and have been used in clinical diagnosis, indicating that there are still numerous bottlenecks to be overcome before magnetic nanoparticles can be practically applied in the biomedical field—examples of these challenges range from the selection of ceramic materials and the synthesis of magnetic nanoparticles to the functionalization of nanoparticle surfaces. In addition to the applications that we describe in this paper, new applications of magnetic nanoparticle composite ceramic materials include in vivo cell-specific material calibration, sensing and tracking, and tissue engineering for regulating and accelerating tissue or cell growth. In addition, the integration of magnetic guidance technology, magnetic thermal therapy technology, and MRI monitoring technology is bound to be among the future trends in development.

## Figures and Tables

**Figure 1 pharmaceutics-14-02584-f001:**
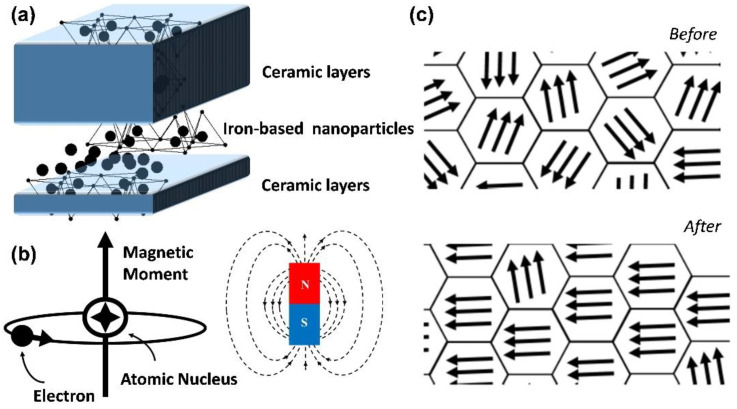
Magnetic variation in iron-based nanoparticles compounded with ceramic materials. Schematic diagram of (**a**) composite material composition; (**b**) magnetic dipole moment of composite material; and (**c**) nonmagnetic/magnetic zone structures.

**Figure 2 pharmaceutics-14-02584-f002:**
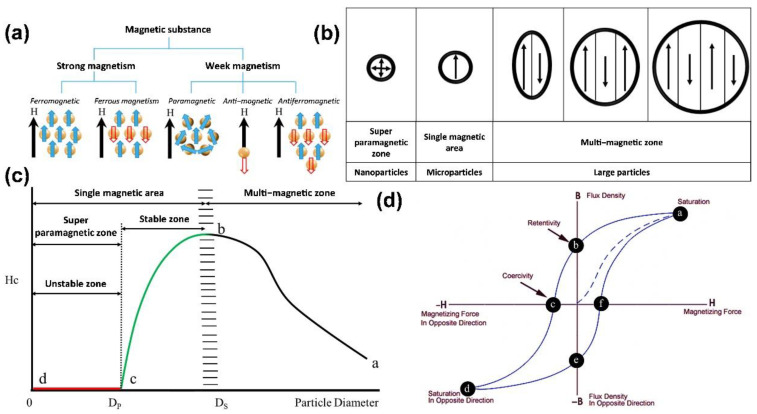
Theories and principles of nanometer sizing and superparamagnetism: (**a**) classification of magnetic substances; (**b**) schematic diagrams of structures of multimagnetic/single-magnetic/superparamagnetic regions; (**c**) relationship between particle size and coercivity. The multi-magnetic region is represented by the curve from a to b, where the coercivity of magnetic particles tends to increase, and the coercivity of the material shows a maximum size at Ds; between b and c is the single-magnetic region, where the particles in this size range show stability; from c to d is the superparamagnetic region, where the material shows an unstable state due to high surface activity at the nanoscale. At D_P_, the material is demagnetized by external thermal effects, resulting in zero coercivity (HC = 0); the zone to the left is called superparamagnetic; and (**d**) hysteresis curves.

**Figure 3 pharmaceutics-14-02584-f003:**
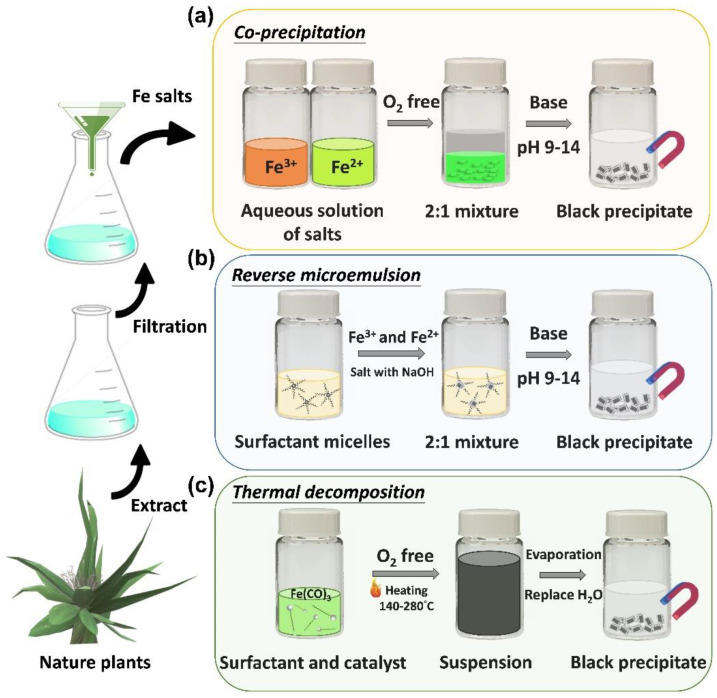
Synthesis of iron oxide nanoparticles by: (**a**) coprecipitation; (**b**) reverse microemulsion; (**c**) thermal decomposition under green synthesis process.

**Figure 4 pharmaceutics-14-02584-f004:**
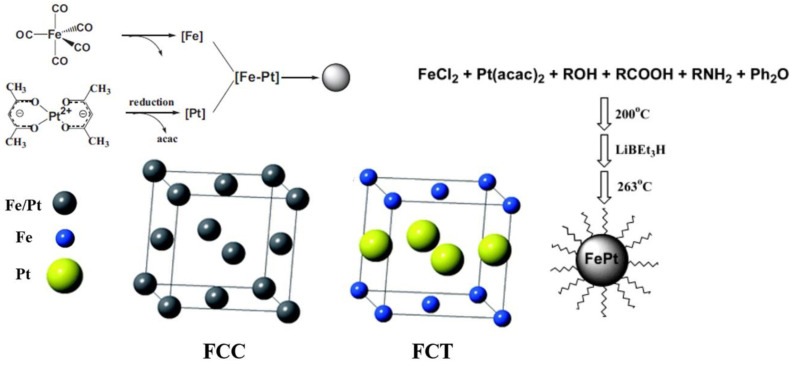
Schematic diagram of synthesizing disordered and ordered structures using Fe(acac)_3_ and Pt(acac)_2_, forming iron–platinum (FePt) nanoparticles from pyrolytic iron and reduced platinum precursors and using ferrous chloride instead of iron pentacarbonyl.

**Figure 5 pharmaceutics-14-02584-f005:**
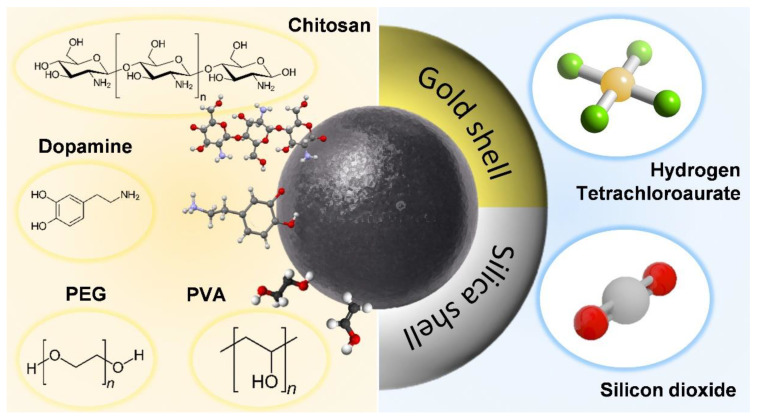
Schematic representation of the surface modification of iron-based nanoparticles. Iron-based nanoparticle with different crosslinker or spacer molecules and layer-by-layer coating. From left to right, the molecules are chitosan, dopamine, polyethylene glycol (PEG), polyvinyl alcohol (PVA), silicon dioxide (SiO_2_), and hydrogen tetrachloroaurate (AuCl_4_).

**Figure 6 pharmaceutics-14-02584-f006:**
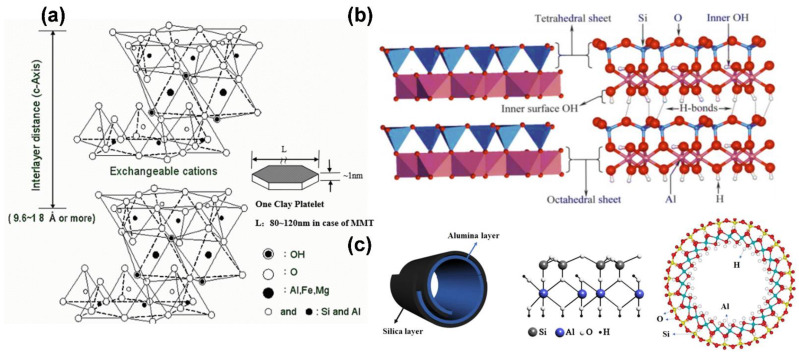
Suitable ceramic carriers of composite iron-based nanomaterials: (**a**) structural diagram of montmorillonite; (**b**) schematic diagram of kaolinite structure; (**c**) schematic diagram of hardystonite structure.

**Figure 7 pharmaceutics-14-02584-f007:**
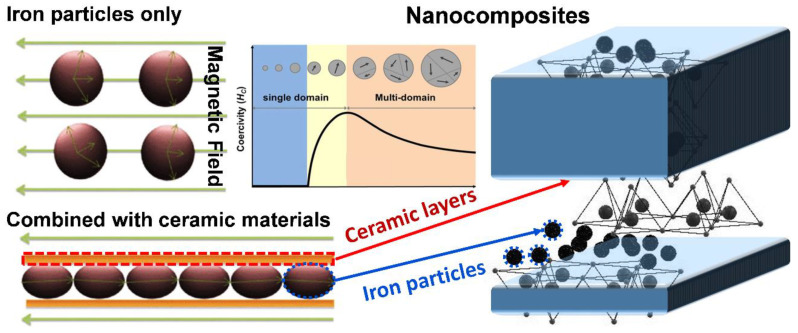
Schematic of possible mechanisms of layer-shaped ceramic materials limiting space of iron-based nanoparticles and with enhanced magnetic properties.

**Figure 8 pharmaceutics-14-02584-f008:**
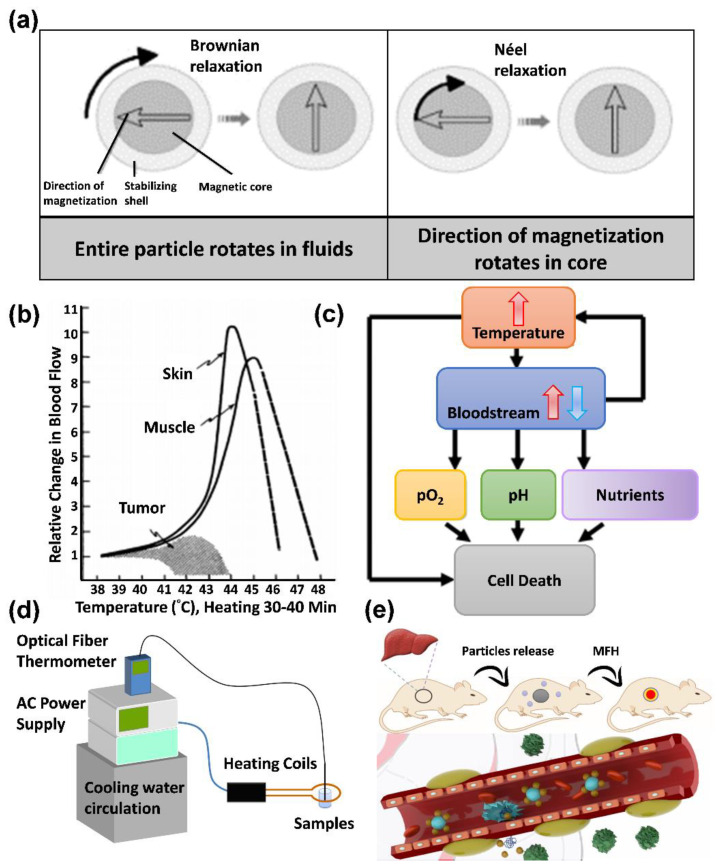
Thermal principles of magnetic fluid hyperthermia and physiological environmental heat therapy: (**a**) Brownian relaxation and Néel relaxation diagrams; (**b**) effect of temperature increase on tumor; (**c**) thermal cell-killing pathway; (**d**) high-frequency magnetic heating induction device; (**e**) iron-based nanoparticle accumulation in tumor tissue based on magnetic guidance, with a higher temperature produced than for normal tissue through the mechanism of enhanced permeability and retention.

**Figure 9 pharmaceutics-14-02584-f009:**
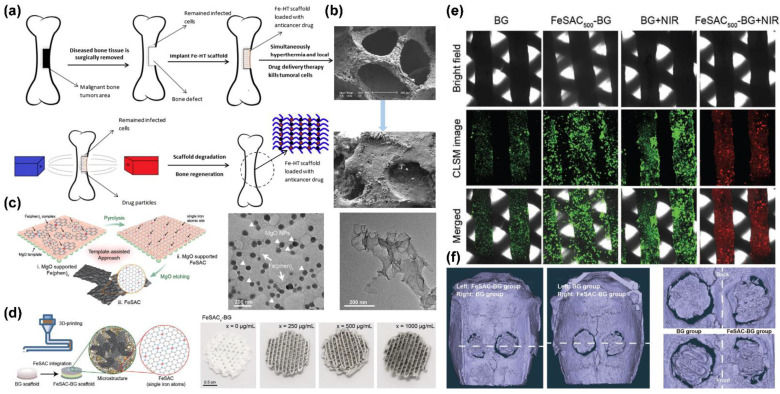
Bone marrow mesenchymal stem cell treatment and cell proliferation during bone cancer treatment: (**a**) schematic diagram of ceramic material containing iron-based nanoparticles bound as biological scaffolds (BGs) after bone cancer surgery; (**b**) BGs immersed in biomimetic body fluid to produce hydroxyapatite; (**c**) 3D mesh ceramic fibers with monocrystalline iron attached to the laminar surface; (**d**) BGs loaded with iron nanoparticles by 3D printing of muffin-like nanocomposite; (**e**) mesenchymal stem cell growth observed under confocal microscope analysis; (**f**) 3D muffin-like BGs embedded in bone to assist bone repair after surgical excision for bone cancer. Adapted with permission from [100,107]. Copyright 2017 Elsevier and 2021 John Wiley and Sons.

**Figure 10 pharmaceutics-14-02584-f010:**
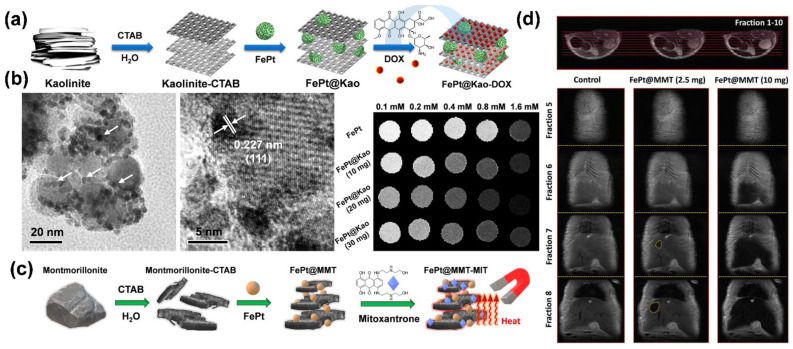
HCC treatment with FePt nanoparticle composite ceramic materials: (**a**) laminated kaolinite structure composite materials; (**b**) sandwich-structured nanoparticles under TEM; (**c**) laminated montmorillonite composite material; (**d**) T2-weighted MRI image of mouse liver tissue. Adapted with permission from Refs. [105,108]. Copyright 2020 ACS Publications and 2021 Springer Nature.

**Table 1 pharmaceutics-14-02584-t001:** MFH bioapplications of ceramic composite iron-based nanomaterials.

Iron-Based Material	Ceramic	Cell Type	Biological Effect	Material Effect	Year	Ref.
Calcium zinc iron silicon oxide composite	Glass	Bone cancer	Promotes osteoblast proliferation	Supports nascent cell proliferation	2011	[96]
Fe/mesoporous bioactive glass	Glass	Human bone marrow mesenchymal stem cells	Improves local delivery of drug therapy and killing of infected tissue cells	Intensifies magnetization	2011	[97]
(Fe^2+^/Fe^3+^)-doped hydroxyapatite	Hydroxyapatite	Osteoblast	Lower level of cytotoxicity achieved	Intensifies magnetization	2012	[98]
Fe_3_O_4_	Magnetic calcium phosphate cement	Breast cancer	Reduces tumor volume	Controlled timing of drug release	2016	[99]
Fe^3+^	Hardystonite	Bone cancer	Enhances drug delivery and killing of tumor cells	Intensifies magnetization	2017	[100]
Ferrimagnetic	Glass	Fibroblast/bone cancer	Does not substantially affect cell morphology	Supports nascent cell proliferation	2017	[101]
Fe_3_O_4_	Hydroxypropyl methylcellulose	Breast cancer	Reduces tumor volume	Controlled timing of drug release	2017	[102]
Fe_3_O_4_	Akermanite	Osteosarcoma	Lower level of cytotoxicity achieved	Controlled timing of drug release	2019	[103]
Magnetic nanoparticles	Calcium phosphate	Mesenchymal stem cell	Increases metabolic activity and proliferation	Intensifies magnetization	2020	[104]
FePt	Kaolinite	Hepatocellular carcinoma	Enhances magnetic signal and killing of tumor cells	Intensifies magnetization	2020	[105]
Hematite nanocrystal	Glass	Fibroblast	Lower level of cytotoxicity achieved	Intensifies magnetization	2021	[106]
Single-atomic iron catalysts	Glass	Bone marrow mesenchymal stem cell	Efficacious osteosarcoma ablation	Supports nascent cell proliferation	2021	[107]
FePt	Montmorillonite	Hepatocellular carcinoma	Enhances magnetic signal and killing of tumor cells	Intensifies magnetization	2021	[108]
Superparamagnetic iron oxide nanoparticles	Glass	Mesenchymal stem cells	Does not affect cell proliferation	Intensifies magnetization	2022	[109]

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
