# Peer review of "Iron-Based Ceramic Composite Nanomaterials for Magnetic Fluid Hyperthermia and Drug Delivery"

_pharmaceutics, 2022, doi:10.3390/pharmaceutics14122584_

Round 1

Reviewer 1 Report

The manuscript entitled Iron-Based Ceramic Composite Nanomaterials for the Applications of Magnetic Fluid Hyperthermia and Drug Delivery tries to give an idea of the potential use of iron-based nanoparticles in two of the most established research areas of nanomedicine: magnetic hyperthermia and drug delivery.

However, to my opinion, provided information is scarce, no classification between iron-based particle is showed and has severe lacks in many important aspects. Therefore, I do not recommend the manuscript for publication.

Among others, I may highlight the following reasons for rejection:

The manuscript lacks enough novelty to be published and is not up to date. For instance, almost the half (49/100) of references have more than 10 years and only one quarter of them later from 2017 (25/100).

There is not a clear relationship with the aim of the journal since very few examples deal with the preparation of nanodevices interesting for pharmacy. For example:

1. From line 109 to 141 the authors extensively explain the fundamentals of magnetization without giving any hint of the limitations or characteristics that are necessary for the therapeutic application of iron-based nanoparticles.

2. In section 2.2 no application of nanomedicine is discussed.

3. In section 3, named Iron-based nanocomposite particles combined with ceramic materials only raw ceramic materials are discussed, without any example bearing iron within their structure.

4. Along breast cancer section (5.1.3.) only discuss generalities of the disease and no relevant examples are given.

Along the manuscript no discussion is made on surface modification necessary for the in vivo application of Fe-containing particles, no discussion is made on how Fe-based particles may be used for Magnetic Resonance Imaging (topic recurrently showed in provided figures) and to conclude, the revision lacks from relevant examples of anticancer drug delivery.

Author Response

To my opinion, provided information is scarce, no classification between iron-based particle is showed and has severe lacks in many important aspects.

Answer: We thank the reviewer for the time taken to review our work and for providing us with critical comments. Here, we will try our best to revise the manuscript to improve the quality under your kind guidance.

The manuscript lacks enough novelty to be published and is not up to date. For instance, almost the half (49/100) of references have more than 10 years and only one quarter of

them later from 2017 (25/100).

Answer: We thank the reviewer for bringing up this important point, and we have updated the references in the manuscript. We cited more references that have been published in recent years to ensure that the novelty fits this Special Issue. Please kindly check the revised references highlighted in red.

References

  1. Vurro, F.; Gerosa, M.; Busato, A.; Muccilli, M.; Milan, E.; Gaudet, J.; Goodwill, P.; Mansfield, J.; Negri, A.; Gherlinzoni, F., et al. Doped Ferrite Nanoparticles Exhibiting Self-Regulating Temperature as Magnetic Fluid Hyperthermia Antitumoral Agents, with Diagnostic Capability in Magnetic Resonance Imaging and Magnetic Particle Imaging. Cancers 2022, 14.
  2. Brollo, M.E.F.; Pinheiro, I.F.; Bassani, G.S.; Varet, G.; Guersoni, V.C.B.; Knobel, M.; Bannwart, A.C.; Muraca, D.; van der Geest, C. Iron Oxide Nanoparticles in a Dynamic Flux: Implications for Magnetic Hyperthermia-Controlled Fluid Viscosity. Acs Appl Nano Mater 2021, 4, 13633-13642.
  3. Limthin, D.; Leepheng, P.; Klamchuen, A.; Phromyothin, D. Enhancement of Electrochemical Detection of Gluten with Surface Modification Based on Molecularly Imprinted Polymers Combined with Superparamagnetic Iron Oxide Nanoparticles. Polymers-Basel 2022, 14.
  4. Khani, T.; Alamzadeh, Z.; Sarikhani, A.; Mousavi, M.; Mirrahimi, M.; Tabei, M.; Irajirad, R.; Abed, Z.; Beik, J. Fe3O4@Au core-shell hybrid nanocomposite for MRI-guided magnetic targeted photo-chemotherapy. Laser Med Sci 2022, 37, 2387-2395.
  5. Abdollahi, B.B.; Ghorbani, M.; Hamishehkar, H.; Malekzadeh, R.; Farajollahi, A.R. Synthesis and characterization of actively HER-2 Targeted Fe3O4@Au nanoparticles for molecular radiosensitization of breast cancer. Bioimpacts 2022, 10.34172/bi.2022.23682.
  6. Le, T.T.; Nguyen, T.N.L.; Nguyen, H.D.; Phan, T.H.T.; Pham, H.N.; Le, D.G.; Hoang, T.P.; Nguyen, T.Q.H.; Le, T.L.; Tran, L.D. Multimodal Imaging Contrast Property of Nano Hybrid Fe3O4@Ag Fabricated by Seed-Growth for Medicinal Diagnosis. Chemistryselect 2022, 7.
  7. Colbert, C.M.; Ming, Z.Y.; Pogosyan, A.; Finn, J.P.; Nguyen, K.L. Comparison of Three Ultrasmall, Superparamagnetic Iron Oxide Nanoparticles for MRI at 3.0 T. J Magn Reson Imaging 2022, 10.1002/jmri.28457.
  8. Meloni, A.; Pistoia, L.; Restaino, G.; Missere, M.; Positano, V.; Spasiano, A.; Casini, T.; Cossu, A.; Cuccia, L.; Massa, A., et al. Quantitative T2*MRI for bone marrow iron overload: normal reference values and assessment in thalassemia major patients. Radiol Med 2022, 127, 1199-1208.
  9. McKiernan, E.P.; Moloney, C.; Chaudhuri, T.R.; Clerkin, S.; Behan, K.; Straubinger, R.M.; Crean, J.; Brougham, D.F. Formation of hydrated PEG layers on magnetic iron oxide nanoflowers shows internal magnetisation dynamics and generates high in-vivo efficacy for MRI and magnetic hyperthermia. Acta Biomater 2022, 152, 393-405.
  10. Chen, B.W.; Chiu, G.W.; He, Y.C.; Huang, C.Y.; Huang, H.T.; Sung, S.Y.; Hsieh, C.L.; Chang, W.C.; Hsu, M.S.; Wei, Z.H., et al. Extracellular and intracellular intermittent magnetic-fluid hyperthermia treatment of SK-Hep1 hepatocellular carcinoma cells based on magnetic nanoparticles coated with polystyrene sulfonic acid. Plos One 2021, 16.
  11. Suleman, M.; Riaz, S. 3D in silico study of magnetic fluid hyperthermia of breast tumor using Fe3O4 magnetic nanoparticles. J Therm Biol 2020, 91.

  1. From line 109 to 141 the authors extensively explain the fundamentals of magnetization without giving any hint of the limitations or characteristics that are necessary for the therapeutic application of iron-based nanoparticles.

Answer: We appreciate the reviewer for raising these comments. In the revised manuscript, we have substantially changed the content of “2. Magnetic properties of nanoparticles.” From line 83 to line 125, we provide more information on the magnetic properties and therapeutic application of iron-based nanoparticles.

As a result, we applied the following corrections:

…Because of the unique properties of magnetic nanoparticles, such as superparamagnetism, high saturation magnetization, and high effective surface area, they are mainly used as contrast agents, to improve image contrast, and as carriers for drug delivery in disease treatment. In addition, when injected into the body, magnetic nanoparticles can generate heat energy through the use of an applied magnetic field to kill cancer cells, avoiding the damage to normal cells observed in conventional chemotherapy and inhibiting cancer cell growth by MFH. Because of their excellent magnetic properties, the application of nanomaterials is constantly being improved and refined [15-17]. All substances have a certain degree of magnetization, which is usually dependent on the material’s atomic structure and surrounding temperature, and the magnetic susceptibility (χ) can be used to express the difficulty of magnetization. When a material is placed under an applied magnetic field (H), its magnetization (M) will change, and the relationship between the two is as follows (Figure 1b):

 … (Equation 2-1)

A magnetic material has a “magnetic domain”, in which the crystalline structure of the energy state itself is divided into several different regions, and these magnetic domains are all oriented in the same direction. Nevertheless, the order of each magnetic domain is not necessarily the same, and there will be mutual offset, as shown in Figure 1c. Assuming that the net magnetic moment is precisely zero, the material is not magnetic. Conversely, when the net magnetic moment is not zero, the material is magnetic [18]. Most materials have the property of being weakly magnetic, even in the absence of an applied magnetic field. The former has an approximate magnetization rate ranging from 10-6 to 10-1 in order of magnitude, while the latter only ranges from 10-6 to 10-3 in order of magnitude. In contrast, some materials can exhibit highly magnetic properties under the action of weak magnetic fields, or even without the application of magnetic fields, such as in the case of ferromagnetic, ferrimagnetic, and antiferromagnetic materials (Figure 2a). In such cases, only a minimal magnetic field is needed to saturate magnetization, and the representative materials are iron, cobalt, and nickel.

Figure 2. Theories and principles of nanometer sizing and superparamagnetism: (a) classification of magnetic substances; (b) schematic diagrams of structures of multimagnetic/single-magnetic/superparamagnetic regions; (c) relationship between particle size and coercivity; and (d) hysteresis curves.

From a microscopic point of view, a large-size ferromagnetic material with multiple magnetic regions exhibits a minor hysteresis effect, as shown in Figure 2b. When the size of the material is reduced to a single domain (generally nanoscale; the critical size varies from material to fabric), the hysteresis effect is the largest (i.e., it has the most substantial coercive force). However, when the scope continues to shrink, the coercive force decreases to zero (i.e., no hysteresis occurs). With the change in the external magnetic field, the data points obtained from the magnetization process of the superparamagnetic nanoparticles can form a hysteresis curve, as shown in Figure 2d. Therefore, the superparamagnetic nanoparticles are not magnetic at room temperature without the applied magnetic field, and when the applied magnetic field is removed, the material’s magnetic properties immediately disappear. For example, in iron oxide nanoparticles usually need to reach a size of several nanometers to become superparamagnetic, as shown in Figure 2c [19].

Please kindly check the revised text of Section 2, highlighted in red.

  1. In section 2.2 no application of nanomedicine is discussed.

Answer: We thank the reviewer for pointing this out. Here, we now include discussion on the application of iron–platinum (FePt) nanoparticles in nanomedicine. We try to point out that FePt nanoparticles have a unique composition and can be applied in MRI and CT.

As a result, we made the following corrections:

…Besides having superior superparamagnetic properties, FePt nanoparticles also have high absorption coefficients for X-rays (Pt absorption coefficient at 50 keV: 6.95 cm2/g). Chou et al. injected 12 nm FePt nanoparticles into mice with tumors by tail-intravenous injection, and the contrast between the MRI and computed tomography (CT) images was substantially improved, which indicates that we can use FePt to track the location of the material in two diagnostic MRI and CT imaging modalities to detect the area in which the MFH and drug release are taking place.

Please kindly check the revised section “Properties of iron-platinum nanoparticles” section, with changes highlighted in red.

  1. In section 3, named Iron-based nanocomposite particles combined with ceramic materialsonly raw ceramic materials are discussed, without any example bearing iron within their structure.

Answer: We thank the reviewer for emphasizing this point. In Paragraph 3, we provide examples of the different types of ceramics materials combined with iron-based nanomaterials. In addition, we also describe how the composite material can be used to help with and improve biomedical applications.

As a result, we made the following corrections:

4.1. Glass (Microscale bioscaffolds)

…In the latest technique, Zhang et al. fabricated a composite scaffold containing mesoporous bioactive glass to encapsulate magnetic Fe3O4 nanoparticles by 3D-printing technology. According to the results, the MBG scaffold structure comprises uniformly sized 400 μm macropores. The magnetic Fe3O4 nanoparticles can be incorporated into the scaffold without affecting its hydroxyapatite mineralization ability while endowing it with excellent magnetic heating ability. In addition, the pore structure can be loaded with doxorubicin (DOX), which is an anticancer drug, and it can thus be used for local drug delivery therapy. The 3D-printed Fe3O4/MBG scaffold shows potential versatility for enhanced osteogenic activity, local anticancer drug delivery, and magnetothermal therapy [71].

4.2. Basic structural nanolayer ceramics

…As previously mentioned, kaolinite has a stable chemical structure, uniformly distributed pore structure, and high adsorption capacity, and it can adsorb different substances in its layered structure, such as FePt nanoparticles or chemotherapeutic drugs. With the adsorption effect provided by kaolinite, FePt nanoparticles can be highly concentrated in a specific space. The nanoparticles can effectively accumulate in a magnetic-field environment according to the influence of the magnetic force to achieve a magnetically controlled MRI effect. If magnetic control is used to guide the accumulation of drugs into tumor tissue, then the side effects caused by chemotherapeutic drugs can be substantially reduced. Currently, a single treatment is not enough to achieve the substantial inhibition of tumor tissue. Cocktail-style therapies have become standard in current cancer treatments. FePt nanoparticles have excellent magnetocaloric effects, and when combined with kaolinite, their heating capacity can be substantially increased. The temperature can be increased to nearly 50 °C; thus, hepatocellular cancer cells can be killed using MFH. At the same time, if kaolinite is loaded with chemotherapeutic drugs, such as Dox, then the system can further inhibit cell growth at the tumor center.

4.3. Basic structure of nanotube ceramics

…In most of the current tubular materials that are compounded with iron-based nanoparticles, carbon nanotubes are used as the carrier, and the hardystonite or akermanite structure is coated with a layer to multiply the nanoparticles. Alternatively, the iron-based nanoparticles can be doped with alpha-Al2O3 crystal to affect carbon nanotube growth in polycrystalline ceramics. Celik et al. prepared Fe-doped Al2O3 ceramics of different textures through templated grain growth and synthesized them into carbon nanotubes via catalytic chemical vapor deposition [75]. According to the experimental results, this novel nanocomposite material has the potential to be used for future biomedical diagnostic and drug delivery applications.

Please kindly check the revised section “Iron-based nanocomposite particles combined with ceramic materials” highlighted in red.

  1. Along breast cancer section (5.1.3.) only discuss generalities of the disease and no relevant examples are given.

Answer: We thank the reviewer for these critical comments. In Section 7.1.3, we provide relevant examples to support the treatment idea.

As a result, we made the following corrections:

...Wang et al. developed a heat-shrinkable, injectable biodegradable material composed of hydroxypropyl methylcellulose (HPMC), polyvinyl alcohol (PVA), and Fe3O4. The authors chose MB-231 for the in vitro experiments to show that the ablation of tumors is positively correlated with the weight of the HPMC/Fe3O4, iron content, and heating time. This novel, safe, and biodegradable material will facilitate the technological transformation of MFH, and it is also expected to introduce new concepts to the field of biomaterial research. Moreover, Tseng et al. used hydroxyapatite (HAP) as a drug carrier for breast cancer treatment via MFH and chemotherapy. The authors developed bifunctional nanoparticles (Pt–Fe-HAP) made of HAP containing iron and platinum ions for combination therapy [120].

Please check the revised section “7.1.3. Breast cancer”, with changes highlighted in red.

Along the manuscript no discussion is made on surface modification necessary for the in vivo application of Fe-containing particles, no discussion is made on how Fe-based particles may be used for Magnetic Resonance Imaging (topic recurrently showed in provided figures) and to conclude, the revision lacks from relevant examples of anti-cancer drug delivery.

Answer: We thank the reviewer for these critical comments. We prepared a new paragraph to discuss the surface modification of Fe-containing particles in “3. Surface modification of iron-based nanoparticles”. In this section, we discuss two methods of particle modification: (1) The use of crosslinker or spacer molecules to form covalent connections, and (2) layer-by-layer coating. Moreover, we added an additional section for MRI diagnosis with ceramic material composite iron nanoparticles.

As a result, we made the following corrections:

  1. Surface modification of iron-based nanoparticles

The key to the technology is how to use ligands for surface modification and increase the function of magnetic nanoparticles. Generally, two methods are used: (1) Crosslinkers or spacer molecules as well as polymer ligands are used to form covalent connections [38]. The body is modified on the surface of the magnetic nanoparticles to include iron nanoparticles as the core and ligands as the shell [39]. The affinity between nanoparticles and polymer ligands depends on the type and quantity of the ligands on the surfaces of the nanoparticles; thus, how to make and select the surface ligands for linking is important [40]. Amines, carboxylates, hydroxyl groups, and thiol groups are commonly used as ligands [41]. In some cases, additional spacer molecules or cross-linking agents are required to facilitate bonding of the nanocomposites [42]; (2) Layer-by-layer coating [43]. With magnetic nanoparticles as the core, other materials are coated, layer by layer, around the nanoparticles based on the electrostatic attraction between opposingcharges [44]. The advantages include the ability to fabricate a single-layer structure and adjust the thickness of the functional shell [45]. According to the above two approaches, we take the carboxylation of chitosan to covalently bond to the surfaces and core–shell structures of the nanoparticles as an example, and we discuss the advantages of ceramic materials for modification of iron-based nanoparticles in the next section (Figure 5).

Chitosans are natural polysaccharides with hydrophilic, biocompatible, biodegradable, and antibacterial properties. They have a good affinity for many biomolecules, which makes them suitable for various biomedical and biotechnology applications. Degradable polymers are more commonly used for the controlled release of drugs [46]. Polysaccharides are nontoxic and biodegradable natural polymers that form particles to coat drugs in acidic environments, such as in the stomach, where they act as antacids to prevent acid damage to drugs. Therefore, they are an ideal material for drug-release-control systems. For drug-targeting applications, magnetic nanoparticles modified with chitosans can adsorb the anticancer drug epirubicin, which indicates a strong interaction between chitosans and epirubicin. In epirubicin-adsorption experiments, the equilibration time is only a few minutes, which means that there is no intrapore diffusion resistance in the adsorption process. Through regulation of the acidic environment in cancer cells at a pH of 4, chitosan is subjected to disintegration. Epirubicin adsorbed on nanomagnetic carriers is expected to be released in in vivo experiments to achieve therapeutic cancer effects [47].

Figure 5. Schematic representation of the surface modification of iron-based nanoparticles. Iron-based nanoparticle with different crosslinker or spacer molecules and layer-by-layer coating. From left to right, the molecules are chitosan, dopamine, polyethylene glycol (PEG), polyvinyl alcohol (PVA), silicon dioxide (SiO2), and hydrogen tetrachloroaurate (AuCl4).

The surface modification of iron-based nanoparticles with core–shell structures is close to that of iron-based nanocomposite particles combined with ceramic materials, which is the focus of this review: i.e., enhancement of the applicability of iron nanoparticles by incorporating other materials. Here, we search for an example of self-assembled nanocomposite materials for iron core–gold shells to link the advantages of ceramic materials combined with iron nanoparticles [48]. The iron core–gold shell composite nanoparticles are selectively toxic to cancer cells. Still, after being placed in water or air for a suitable period, they are no longer harmful to cancer cells [49]. Researchers found that freshly produced iron core–gold shell composite nanoparticles are not toxic to cancer cells when placed in water. Water molecules will penetrate through the grain interface of the gold shell and react with iron at the gold–iron interface to produce ferrous ions, which are gradually released to kill cancer cells. However, the dissolved oxygen in the water will also spread to the gold–iron interface through the grain interface of the gold shell, oxidizing the iron into iron oxide, which forms a protective layer to prevent the continued production and release of ferrous ions and, thus, no longer having a toxic killing function and achieving the effect of self-liquidation [50]. Likewise, protection is also provided by the ceramic material compounded with iron-based nanoparticles. In addition, due to its porous nature, the ceramic material can also offer drug loading and delivery of iron nanoparticles, similarly to chitosan mentioned above.

As a result, the following corrections have been made:

  1. Magnetic resonance imaging (MRI) with ceramic material composite iron nanoparticles

Among the many screening and diagnostic methods, magnetic resonance imaging (MRI) can provide high-resolution images of the liver without the need for ionizing radiation. Consequently, MRI is the best choice for initial tumor diagnosis. For this reason, iron-based nanomaterials have become candidates for MRI imaging because of their excellent T2 contrast ability [76]. However, most of the commonly used magnetic vibrating carriers currently available in the market are iron oxide particles or strontium ion complexes, which may cause side effects, such as nausea, allergic reactions, and kidney injury [77]. To address this, the goal of current research is to improve the performance of magnetic carriers through development of a multifunctional composite nanocarrier that can be used for high-resolution MRI with low toxicity and a therapeutic effect [78].

Figure 7. Schematic of possible mechanisms of layer-shaped ceramic materials limiting space of iron-based nanoparticles and with enhanced magnetic properties.

Positively charged atomic nuclei spin in random environments. In this case, the nuclear spin axes are arranged in a random pattern, and when placed in a static magnetic field, the nucleus spins in the direction of the applied magnetic field. The spin frequency is called the Larmor frequency, and it is related to the properties of the nucleus itself and is proportional to the strength of the applied magnetic field. The effect is that the iron-based nanoparticles are subjected to single-axial compressive stress, and the mechanical stress causes the rearrangement of the magnetic dipoles of the iron-based nanoparticles and places them parallel with the direction of the ceramic layer. Therefore, in human MRI, the degree of saturation magnetization along the magnetization direction depends on the hydrogen atoms, which are the main source of nuclei because the human body is mainly composed of water, and it is the hydrogen atoms in water that help in visualizing the image. As the magnetization vector of the atomic nucleus gradually increases and then stabilizes during the spinning process, the nucleus will resonate if disturbed by a fixed-frequency radio frequency (Figure 7). The resonance effect, which is limited by the ceramic material, brings higher saturation magnetization to the iron-based nanoparticles, thereby enhancing their ability for T2-weighted MRI diagnostic imaging.

Please kindly check the revised Sections 3 and 5, with changes highlighted in red.

Reviewer 2 Report

An interesting review that presents the application of Iron-Based Ceramic Composite Nanomaterials for Hyperthermia and Drug Delivery. The manuscript is well-organized and clearly presented. I have just some comments:

1-page 5, line 182: face-centered cubic (FCC) and face-centered tetragonal (FCT)

2-Page9, lines 346 and 347: the angular frequency has repeated two times.

3-Page4, line 156: tumors smaller than 7 mm have repeated two times.

4- The size and quality of the figure 7 needs to improved.

Author Response

An interesting review that presents the application of Iron-Based Ceramic Composite Nanomaterials for Hyperthermia and Drug Delivery. The manuscript is well-organized and clearly presented.

Answer: We thank the reviewer for the positive encouragement and kind suggestions. Based on the reviewer’s recommendations, we have made corrections to the manuscript. Please refer to the manuscript for the corrections and additions of phrases highlighted in red.

1-page 5, line 182: face-centered cubic (FCC) and face-centered tetragonal (FCT)

Answer: The abbreviations of FCC and FCT have been added to the manuscript.

2-Page9, lines 346 and 347: the angular frequency has repeated two times.

Answer: We have removed the inadvertent repetition of “the angular frequency” from the manuscript.

3-Page4, line 156: tumors smaller than 7 mm have repeated two times.

Answer: We have removed the inadvertent repetition of “tumors smaller than 7 mm” from the manuscript.

4- The size and quality of the figure 7 needs to improved.

Answer: We have improved the size and quality of Figure 7.

Reviewer 3 Report

The review by Chan et al. attempts to cover recent advances in the exciting field of iron-based composite nanomaterials for hyperthermia and drug delivery. Literature analysis is reasonable, considering the amount of articles published on the topic. However ,the review largely fails in providing meaningful insights in novel developments in the field. Main criticisms that should be addressed are:

1) Background information on base concepts on nanomaterials and magnetism should be avoided. This information is, in this review, forcedly approximate an limited ,and does not add anything to the general knowledge of the average reader.

2) The authors chose to divide their review in different sections dealing with several pathologies (e.g. different tumors), rather than a more logical and linear presentation of application in hyperthermia, drug delivery, and diagnosis. The authors should consider revision of the review from this different perspective.

3) Although diffused in scientific literature, drug delivery fo chemiotherapy with nanoparticles is not a clinically realistic strategy. The authors should clearly state this, and possibly concentrate on those fields where magnetic nanoparticles can have an edge (diagnosis and hyperthermia). I think that other approaches for drug delivery are so limited that they should not be considered.

4) Last, but not least, the review is really difficult to read due to an unacceptably low English level. Please check carefully all the manuscript and have it checked by someone with a good English knowledge.

Author Response

The review by Chan et al. attempts to cover recent advances in the exciting field of iron-based composite nanomaterials for hyperthermia and drug delivery. Literature analysis is reasonable, considering the amount of articles published on the topic. However ,the review largely fails in providing meaningful insights in novel developments in the field. Main criticisms that should be addressed are:

Answer: We are thankful for the reviewer’s kind concerns. Based on the reviewer’s suggestions, we have made corrections to the manuscript. Here, we revised the manuscript to improve the quality with your kind guidance.

1) Background information on base concepts on nanomaterials and magnetism should be avoided. This information is, in this review, forcedly approximate an limited ,and does not add anything to the general knowledge of the average reader.

Answer: We appreciate the reviewer for raising the comments. In the revised manuscript, we have substantially changed the content of “2. Magnetic properties of nanoparticles.” From line 83 to line 125, we provide more information on the magnetic properties and therapeutic application of iron-based nanoparticles.

As a result, the following corrections have been made:

…Because of the unique properties of magnetic nanoparticles, such as superparamagnetism, high saturation magnetization, and high effective surface area, they are mainly used as contrast agents, to improve image contrast, and as carriers for drug delivery in disease treatment. In addition, when injected into the body, magnetic nanoparticles can generate heat energy through the use of an applied magnetic field to kill cancer cells, avoiding the damage to normal cells observed in conventional chemotherapy and inhibiting cancer cell growth by MFH. Because of their excellent magnetic properties, the application of nanomaterials is constantly being improved and refined [15-17]. All substances have a certain degree of magnetization, which is usually dependent on the material’s atomic structure and surrounding temperature, and the magnetic susceptibility (χ) can be used to express the difficulty of magnetization. When a material is placed under an applied magnetic field (H), its magnetization (M) will change, and the relationship between the two is as follows (Figure 1b):

 … (Equation 2-1)

A magnetic material has a “magnetic domain”, in which the crystalline structure of the energy state itself is divided into several different regions, and these magnetic domains are all oriented in the same direction. Nevertheless, the order of each magnetic domain is not necessarily the same, and there will be mutual offset, as shown in Figure 1c. Assuming that the net magnetic moment is precisely zero, the material is not magnetic. Conversely, when the net magnetic moment is not zero, the material is magnetic [18]. Most materials have the property of being weakly magnetic, even in the absence of an applied magnetic field. The former has an approximate magnetization rate ranging from 10-6 to 10-1 in order of magnitude, while the latter only ranges from 10-6 to 10-3 in order of magnitude. In contrast, some materials can exhibit highly magnetic properties under the action of weak magnetic fields, or even without the application of magnetic fields, such as in the case of ferromagnetic, ferrimagnetic, and antiferromagnetic materials (Figure 2a). In such cases, only a minimal magnetic field is needed to saturate magnetization, and the representative materials are iron, cobalt, and nickel.

Figure 2. Theories and principles of nanometer sizing and superparamagnetism: (a) classification of magnetic substances; (b) schematic diagrams of structures of multimagnetic/single-magnetic/superparamagnetic regions; (c) relationship between particle size and coercivity; and (d) hysteresis curves.

From a microscopic point of view, a large-size ferromagnetic material with multiple magnetic regions exhibits a minor hysteresis effect, as shown in Figure 2b. When the size of the material is reduced to a single domain (generally nanoscale; the critical size varies from material to fabric), the hysteresis effect is the largest (i.e., it has the most substantial coercive force). However, when the scope continues to shrink, the coercive force decreases to zero (i.e., no hysteresis occurs). With the change in the external magnetic field, the data points obtained from the magnetization process of the superparamagnetic nanoparticles can form a hysteresis curve, as shown in Figure 2d. Therefore, the superparamagnetic nanoparticles are not magnetic at room temperature without the applied magnetic field, and when the applied magnetic field is removed, the material’s magnetic properties immediately disappear. For example, in iron oxide nanoparticles usually need to reach a size of several nanometers to become superparamagnetic, as shown in Figure 2c [19].

Please kindly check the revised Section 2, with changes highlighted in red.

2) The authors chose to divide their review in different sections dealing with several pathologies (e.g. different tumors), rather than a more logical and linear presentation of application in hyperthermia, drug delivery, and diagnosis. The authors should consider revision of the review from this different perspective.

Answer: We thank the reviewer for pointing this out. We have reorganized the manuscript headings and divided the discussions on MRI (in section 5), MFH (in section 6), and drug delivery (in section 7) into three paragraphs. The most critical new paragraph is the analysis of MRI diagnosis.

As a result, the following corrections have been made:

  1. Magnetic resonance imaging (MRI) with ceramic material composite iron nanoparticles

Among the many screening and diagnostic methods, magnetic resonance imaging (MRI) can provide high-resolution images of the liver without the need for ionizing radiation. Consequently, MRI is the best choice for initial tumor diagnosis. For this reason, iron-based nanomaterials have become candidates for MRI imaging because of their excellent T2 contrast ability [76]. However, most of the commonly used magnetic vibrating carriers currently available in the market are iron oxide particles or strontium ion complexes, which may cause side effects, such as nausea, allergic reactions, and kidney injury [77]. To address this, the goal of current research is to improve the performance of magnetic carriers through development of a multifunctional composite nanocarrier that can be used for high-resolution MRI with low toxicity and a therapeutic effect [78].

Figure 7. Schematic of possible mechanisms of layer-shaped ceramic materials limiting space of iron-based nanoparticles and with enhanced magnetic properties.

Positively charged atomic nuclei spin in random environments. In this case, the nuclear spin axes are arranged in a random pattern, and when placed in a static magnetic field, the nucleus spins in the direction of the applied magnetic field. The spin frequency is called the Larmor frequency, and it is related to the properties of the nucleus itself and is proportional to the strength of the applied magnetic field. The effect is that the iron-based nanoparticles are subjected to single-axial compressive stress, and the mechanical stress causes the rearrangement of the magnetic dipoles of the iron-based nanoparticles and places them parallel with the direction of the ceramic layer. Therefore, in human MRI, the degree of saturation magnetization along the magnetization direction depends on the hydrogen atoms, which are the main source of nuclei because the human body is mainly composed of water, and it is the hydrogen atoms in water that help in visualizing the image. As the magnetization vector of the atomic nucleus gradually increases and then stabilizes during the spinning process, the nucleus will resonate if disturbed by a fixed-frequency radio frequency (Figure 7). The resonance effect, which is limited by the ceramic material, brings higher saturation magnetization to the iron-based nanoparticles, thereby enhancing their ability for T2-weighted MRI diagnostic imaging.

Please kindly check the revised Section 5, with changes highlighted in red.

3) Although diffused in scientific literature, drug delivery fo chemiotherapy with nanoparticles is not a clinically realistic strategy. The authors should clearly state this, and possibly concentrate on those fields where magnetic nanoparticles can have an edge (diagnosis and hyperthermia). I think that other approaches for drug delivery are so limited that they should not be considered.

Answer: We thank the reviewer for the insightful suggestions. We agree that drug delivery with nanocomplexes may not be a clinically appropriate treatment. Therefore, we merged the paragraph on drug delivery with those on the diagnosis (in section 5) and MFH (in section 6). However, because the Special Issue is focused on the drug delivery process, we also present examples of drug delivery methods in several cancer therapies in the hope that the nanocomposite may have clinical application in the future.

4) Last, but not least, the review is really difficult to read due to an unacceptably low English level. Please check carefully all the manuscript and have it checked by someone with a good English knowledge.

Answer: We thank the reviewer for pointing this out. To address this concern, we have corrected several spelling mistakes throughout the manuscript and sent the paper to the MDPI English Editing Service for further proofreading and grammar evaluation to improve the English quality. Please refer to the English certificate (ID: english-edited-53535) for this revision.

Reviewer 4 Report

In this review, application of Iron based nanomaterials in medicine especially the magnetic hyperthermia and drug delivery were reported using the results of various papers in literatures.

In general, the manuscript is written well and the discussion are satisfied. Moreover, the review is useful and valuable for readers. However, there are several point that should properly addressed.

keywords: please capitalize the words.

Introduction and all the text: please insert proper ref for each figure.

I recommend to use the following papers and compare the results of changing the anisotropy on the ferrites properties especially hyperthermia efficiency. In addition, the effect of interparticle interaction on hyperthermia output should well discused:

Beilstein journal of nanotechnology 10 (1), 1348-1359.

Beilstein journal of nanotechnology 10 (1), 856-865.

4. Magnetic fluid hyperthermia (MFH)

Please use results of proper references for equations and add some additional parameters like, concentration, polymer coating, field and frequency effect, aggregation and interparticle interaction in ferrite nanoparticles etc. For this, I recommend to use the results of following papers and any relevant refs in literature:

Journal of Applied Physics 119 (6), 063901

Materials Research Express 4 (7), 075051

Current Applied Physics 12 (3), 812-816

Journal of Magnetism and Magnetic Materials 385, 308-312

Journal of nanoparticle research 15 (2), 1-12

Journal of magnetism and magnetic materials 324 (2), 154-160

Journal of Water and Environmental Nanotechnology 6 (2), 109-120

Physica C: Superconductivity and its applications 549, 119-121

At the last paragraph of this part the authors should provide some reason for using polymer coating beside bio-compatibility in MHT. The coating will affect the interparticle interaction between aggregates and can change the magnetic properties, which directly affect the MFH efficiency. Please use the results of similar works in literature. As example following work are useful to justify the coating:

Journal of Magnetism and Magnetic Materials 399, 236-244

Journal of Materials Science: Materials in Electronics 32 (19), 24026-24040

Ceramics International 48 (19), 27995-28005

5.1.3. Breast cancer

Please add relevant refs in this part. 

Author Response

In general, the manuscript is written well and the discussion are satisfied. Moreover, the review is useful and valuable for readers. However, there are several point that should properly addressed.

Answer: We thank the reviewer for the positive encouragement and kind suggestions. Based on the reviewer’s advice, we have made corrections to the manuscript. Please refer to the manuscript for the corrections and additions of phrases, highlighted in red.

keywords: please capitalize the words.

Answer: We thank the reviewer for this comment. We have capitalized the keywords.

Introduction and all the text: please insert proper ref for each figure.

Answer: We are thankful for your suggestions and recommendations. Figures 1 to 6 were mainly drawn by us, whereas Figure 7 and subsequent figures were reproduced from other reports; thus, we have modified the text accompanying these figures to include references.

I recommend to use the following papers and compare the results of changing the anisotropy on the ferrites properties especially hyperthermia efficiency. In addition, the effect of interparticle interaction on hyperthermia output should well discussed.

Answer: We thank the reviewer for sharing this critical reference with us. We followed the definitions of the anisotropy on the ferrites properties in this article and have updated the references. Please kindly check the revised sentences in the introduction  and references highlighted in red.

References

  1. Aslibeiki, B.; Kameli, P.; Salamati, H.; Concas, G.; Fernandez, M.S.; Talone, A.; Muscas, G.; Peddis, D. Co-doped MnFe2O4 nanoparticles: magnetic anisotropy and interparticle interactions. Beilstein J Nanotech 2019, 10, 856-865.
  2. Jalili, H.; Aslibeiki, B.; Varzaneh, A.G.; Chernenko, V.A. The effect of magneto-crystalline anisotropy on the properties of hard and soft magnetic ferrite nanoparticles. Beilstein J Nanotech 2019, 10, 1348-1359.

  1. Magnetic fluid hyperthermia (MFH)

Please use results of proper references for equations and add some additional parameters like, concentration, polymer coating, field and frequency effect, aggregation and interparticle interaction in ferrite nanoparticles etc.

Answer: We thank the reviewer for sharing this critical reference with us. We followed the definitions of some additional parameters in this article and updated the references in this section. Please kindly check the revised sentences and references highlighted in red.

References

  1. Aslibeiki, B.; Kameli, P.; Manouchehri, I.; Salamati, H. Strongly interacting superspins in Fe3O4 nanoparticles. Curr Appl Phys 2012, 12, 812-816.
  2. Aslibeiki, B.; Kameli, P.; Salamati, H. The role of Ag on dynamics of superspins in MnFe2-xAgxO4 nanoparticles. J Nanopart Res 2013, 15.
  3. Aslibeiki, B.; Ehsani, M.H.; Nasirzadeh, F.; Mohammadi, M.A. The effect of interparticle interactions on spin glass and hyperthermia properties of Fe3O4 nanoparticles. Mater Res Express 2017, 4.
  4. Lotfi, S.; Aslibeiki, B.; Zarei, M. Efficient Pb (II) removal from wastewater by TEG coated Fe3O4 ferrofluid. Journal of Water and Environmental Nanotechnology 2021, 6, 109-120.
  5. Aslibeiki, B.; Kameli, P.; Salamati, H. The effect of grinding on magnetic properties of agglomereted MnFe2O4 nanoparticles. J Magn Magn Mater 2012, 324, 154-160.
  6. Aslibeiki, B.; Kameli, P. Magnetic properties of MnFe2O4 nano-aggregates dispersed in paraffin wax. J Magn Magn Mater 2015, 385, 308-312.
  7. Aslibeiki, B.; Kameli, P.; Salamati, H. The effect of dipole-dipole interactions on coercivity, anisotropy constant, and blocking temperature of MnFe2O4 nanoparticles. J Appl Phys 2016, 119.
  8. Ebrahimisadr, S.; Aslibeiki, B.; Asadi, R. Magnetic hyperthermia properties of iron oxide nanoparticles: The effect of concentration. Physica C 2018, 549, 119-121.

At the last paragraph of this part the authors should provide some reason for using polymer coating beside bio-compatibility in MHT. The coating will affect the interparticle interaction between aggregates and can change the magnetic properties, which directly affect the MFH efficiency. Please use the results of similar works in literature.

Answer: We thank the reviewer for these critical comments. We created a new paragraph to discuss the topic surface modification of Fe-containing particles in the section “3. Surface modification of iron-based nanoparticles”. In this section, we discuss two methods of particle modification: (1) the use of crosslinker or spacer molecules to form covalent connections, and (2) layer-by-layer coating.

As a result, we made the following corrections:

  1. Surface modification of iron-based nanoparticles

The key to the technology is how to use ligands for surface modification and increase the function of magnetic nanoparticles. Generally, two methods are used: (1) Crosslinkers or spacer molecules as well as polymer ligands are used to form covalent connections [38]. The body is modified on the surface of the magnetic nanoparticles to include iron nanoparticles as the core and ligands as the shell [39]. The affinity between nanoparticles and polymer ligands depends on the type and quantity of the ligands on the surfaces of the nanoparticles; thus, how to make and select the surface ligands for linking is important [40]. Amines, carboxylates, hydroxyl groups, and thiol groups are commonly used as ligands [41]. In some cases, additional spacer molecules or cross-linking agents are required to facilitate bonding of the nanocomposites [42]; (2) Layer-by-layer coating [43]. With magnetic nanoparticles as the core, other materials are coated, layer by layer, around the nanoparticles based on the electrostatic attraction between opposingcharges [44]. The advantages include the ability to fabricate a single-layer structure and adjust the thickness of the functional shell [45]. According to the above two approaches, we take the carboxylation of chitosan to covalently bond to the surfaces and core–shell structures of the nanoparticles as an example, and we discuss the advantages of ceramic materials for modification of iron-based nanoparticles in the next section (Figure 5).

Chitosans are natural polysaccharides with hydrophilic, biocompatible, biodegradable, and antibacterial properties. They have a good affinity for many biomolecules, which makes them suitable for various biomedical and biotechnology applications. Degradable polymers are more commonly used for the controlled release of drugs [46]. Polysaccharides are nontoxic and biodegradable natural polymers that form particles to coat drugs in acidic environments, such as in the stomach, where they act as antacids to prevent acid damage to drugs. Therefore, they are an ideal material for drug-release-control systems. For drug-targeting applications, magnetic nanoparticles modified with chitosans can adsorb the anticancer drug epirubicin, which indicates a strong interaction between chitosans and epirubicin. In epirubicin-adsorption experiments, the equilibration time is only a few minutes, which means that there is no intrapore diffusion resistance in the adsorption process. Through regulation of the acidic environment in cancer cells at a pH of 4, chitosan is subjected to disintegration. Epirubicin adsorbed on nanomagnetic carriers is expected to be released in in vivo experiments to achieve therapeutic cancer effects [47].

Figure 5. Schematic representation of the surface modification of iron-based nanoparticles. Iron-based nanoparticle with different crosslinker or spacer molecules and layer-by-layer coating. From left to right, the molecules are chitosan, dopamine, polyethylene glycol (PEG), polyvinyl alcohol (PVA), silicon dioxide (SiO2), and hydrogen tetrachloroaurate (AuCl4).

The surface modification of iron-based nanoparticles with core–shell structures is close to that of iron-based nanocomposite particles combined with ceramic materials, which is the focus of this review: i.e., enhancement of the applicability of iron nanoparticles by incorporating other materials. Here, we search for an example of self-assembled nanocomposite materials for iron core–gold shells to link the advantages of ceramic materials combined with iron nanoparticles [48]. The iron core–gold shell composite nanoparticles are selectively toxic to cancer cells. Still, after being placed in water or air for a suitable period, they are no longer harmful to cancer cells [49]. Researchers found that freshly produced iron core–gold shell composite nanoparticles are not toxic to cancer cells when placed in water. Water molecules will penetrate through the grain interface of the gold shell and react with iron at the gold–iron interface to produce ferrous ions, which are gradually released to kill cancer cells. However, the dissolved oxygen in the water will also spread to the gold–iron interface through the grain interface of the gold shell, oxidizing the iron into iron oxide, which forms a protective layer to prevent the continued production and release of ferrous ions and, thus, no longer having a toxic killing function and achieving the effect of self-liquidation [50]. Likewise, protection is also provided by the ceramic material compounded with iron-based nanoparticles. In addition, due to its porous nature, the ceramic material can also offer drug loading and delivery of iron nanoparticles, similarly to chitosan mentioned above.

Please kindly check the revised Section 3, with changes highlighted in red.

References

  1. Aslibeiki, B.; Eskandarzadeh, N.; Jalili, H.; Varzaneh, A.G.; Kameli, P.; Orue, I.; Chernenko, V.; Hajalilou, A.; Ferreira, L.P.; Cruz, M.M. Magnetic hyperthermia properties of CoFe2O4 nanoparticles: Effect of polymer coating and interparticle interactions. Ceram Int 2022, 48, 27995-28005.
  2. Rezanezhad, A.; Hajalilou, A.; Eslami, F.; Parvini, E.; Abouzari-Lotf, E.; Aslibeiki, B. Superparamagnetic magnetite nanoparticles for cancer cells treatment via magnetic hyperthermia: effect of natural capping agent, particle size and concentration. J Mater Sci-Mater El 2021, 32, 24026-24040.
  3. Aslibeiki, B.; Kameli, P.; Ehsani, M.H.; Salamati, H.; Muscas, G.; Agostinelli, E.; Foglietti, V.; Casciardi, S.; Peddis, D. Solvothermal synthesis of MnFe2O4 nanoparticles: The role of polymer coating on morphology and magnetic properties. J Magn Magn Mater 2016, 399, 236-244.

5.1.3. Breast cancer

Please add relevant refs in this part. 

Answer: We thank the reviewer for these critical comments and the reviewer for sharing this critical reference with us. In Section 7.1.3, we provide relevant examples to support the treatment concept. Please kindly check the revised sentences and references highlighted in red.

As a result, we made the following corrections:

…Wang et al. developed a heat-shrinkable, injectable biodegradable material composed of hydroxypropyl methylcellulose (HPMC), polyvinyl alcohol (PVA), and Fe3O4. The authors chose MB-231 for the in vitro experiments to show that the ablation of tumors is positively correlated with the weight of the HPMC/Fe3O4, iron content, and heating time. This novel, safe, and biodegradable material will facilitate the technological transformation of MFH, and it is also expected to introduce new concepts to the field of biomaterial research. Moreover, Tseng et al. used hydroxyapatite (HAP) as a drug carrier for breast cancer treatment via MFH and chemotherapy. The authors developed bifunctional nanoparticles (Pt–Fe-HAP) made of HAP containing iron and platinum ions for combination therapy [120].

Please check the revised section “7.1.3. Breast cancer”, with changes highlighted in red.

Round 2

Reviewer 1 Report

In this revised version of the manuscript titled Iron-based Ceramic Composite Nanomaterials for Magnetic Fluid Hyperthermia and Drug Delivery the authors have clearly improved the quality of the initial manuscript. The addition of a significant number of very recent references, the coverage of some topics previously overlooked together with the provided a plagiarism report and English editing, has clearly improved readability and potential impact of this work.

Although the topic covered is complex and I still miss some relevant biomedical aspects in the application of reported composite materials (in vivo degradation and residence time, surface functionalization, long-term toxicity, attenuation of magnetic susceptibility and ability to heating in composites, etc.), to my opinion the revision in its actual form could be of interest for material’s scientists with its current structure and could be considered for publication in Pharmaceutics.

However, prior to publication, I would suggest to accomplish some very minor revisions to the manuscript:

In section 4, the text refers to composites with ceramic materials; but among all the ceramic materials known, herein only those based on silicon and aluminum are mentioned (akermanite, kaolinite, hardystonite, montmorillonite, hydroxyapatite, etc.). For this reason, I suggest indicating somehow their enhanced biocompatibility. For instance, I would propose the authors to name bioactive glasses in section 4.1 and biocompatible ceramics in sections 4.2 and 4.3.

I would also appreciate a homogenization in noted dimensions (nanometer vs Angstrom notation in lines 319, 321, 369, and 370) and provide complete details for references 24, 49, 76, 86.

Author Response

Response to Reviewer 1:

Reviewer #1 (Reviewer Comments to the Author):

In this revised version of the manuscript titled Iron-based Ceramic Composite Nanomaterials for Magnetic Fluid Hyperthermia and Drug Delivery the authors have clearly improved the quality of the initial manuscript. The addition of a significant number of very recent references, the coverage of some topics previously overlooked together with the provided a plagiarism report and English editing, has clearly improved readability and potential impact of this work.

Answer: We thank the reviewer for the positive encouragement and kind suggestions. Based on the reviewer’s recommendations, we have made corrections to the manuscript. Please refer to the manuscript for the corrections and additions of phrases highlighted in red.

In section 4, the text refers to composites with ceramic materials; but among all the ceramic materials known, herein only those based on silicon and aluminum are mentioned (akermanite, kaolinite, hardystonite, montmorillonite, hydroxyapatite, etc.). For this reason, I suggest indicating somehow their enhanced biocompatibility. For instance, I would propose the authors to name bioactive glasses in section 4.1 and biocompatible ceramics in sections 4.2 and 4.3.

Answer: We thank the reviewer for the time taken to review our work and for providing us with critical comments. Here, we added one paragraph with the reference citation to roughly discuss ceramic materials’ potential toxicity and biocompatibility at the start of section 4.

As a result, we applied the following corrections:

…As a new type of drug delivery system, ceramic nanocarriers have high mechanical strength, good body response, and low or non-existing biodegradability. Ceramic nanocarriers can protect the drug and the composite nanoparticles from pH and temperature effects. However, despite the high biocompatibility shown in current studies, there is still a lack of information on their clinical use [51]. The research journey for future applications of ceramic nanocarriers is still long; thus, this section will focus on the improvements brought by ceramic materials composites.

References

  1. Singh, D.; Singh, S.; Sahu, J.; Srivastava, S.; Singh, M.R. Ceramic nanoparticles: Recompense, cellular uptake and toxicity concerns. Artif. Cell Nanomed. B 2016, 44, 401-409.

Please kindly check the revised text of Section 4, highlighted in red.

I would also appreciate a homogenization in noted dimensions (nanometer vs Angstrom notation in lines 319, 321, 369, and 370) and provide complete details for references 24, 49, 76, 86.

Answer: We thank the reviewer for bringing up this important point, and we have revised homogenization in the noted dimensions and updated the references in the manuscript. For references 76 and 86, since these two references are the early access articles that might not be suitable for citation, we replace the other two related articles that have been published in the last five years.

As a result, we applied the following corrections:

Line 377… hydrate elite (10 × 10-1 nm)

Line 378… dehydrated elite (7 × 10-1 nm)

References

24. Aslibeiki, B.; Kameli, P.; Salamati, H. The role of Ag on dynamics of superspins in MnFe2-xAgxO4 nanoparticles. J. Nanopart. Res. 2013, 15, 1430.

49. Abdollahi, B.B.; Ghorbani, M.; Hamishehkar, H.; Malekzadeh, R.; Farajollahi, A.R. Synthesis and characterization of actively HER-2 Targeted Fe3O4@Au nanoparticles for molecular radiosensitization of breast cancer. Bioimpacts 2022, 12, 23682.

77. Barcelos, K.A.; Tebaldi, M.L.; do Egito, E.S.T.; Leao, N.M.; Soares, D.C.F. PEG-Iron Oxide Core-Shell Nanoparticles: In situ Synthesis and In vitro Biocompatibility Evaluation for Potential T-2-MRI Applications. Bionanoscience 2020, 10, 1107-1120.

88. Ring, H.L.; Bischof, J.C.; Garwood, M. Use and Safety of Iron Oxide Nanoparticles in MRI and MFH. eMagRes 2019, 8, 265-277.

Please kindly check the revised text of Section 4, highlighted in red.

Reviewer 3 Report

I thank the authors for considering and addressing my concerns. I think the manuscript is now suitable for publication

Author Response

Response to Reviewer 3:

Reviewer #3 (Reviewer Comments to the Author): 

I thank the authors for considering and addressing my concerns. I think the manuscript is now suitable for publication

Answer: We thank the reviewer for the positive encouragement and kind suggestions. This manuscript discusses how ceramics nanocarrier and FDA-approved iron-based nanoparticles can be used in MRI diagnostics, MFH therapy, and drug delivery. In section 4, we add a paragraph on the advantages of ceramics nanocarrier.

As a result, we applied the following corrections:

…As a new type of drug delivery system, ceramic nanocarriers have high mechanical strength, good body response, and low or non-existing biodegradability. Ceramic nanocarriers can protect the drug and the composite nanoparticles from pH and temperature effects. However, despite the high biocompatibility shown in current studies, there is still a lack of information on their clinical use [51]. The research journey for future applications of ceramic nanocarriers is still long; thus, this section will focus on the improvements brought by ceramic materials composites.

Please kindly check the revised text of Section 4, highlighted in red.
